# FLEX-Judge: Text-Only Reasoning Unleashes Zero-Shot Multimodal Evaluators

**Jongwoo Ko**[1][*]   **Sungnyun Kim**[2][*]   **Sungwoo Cho**[2]   **Se-Young Yun**[2]
[1]Microsoft        [2]KAIST AI
jongwooko@microsoft.com, {ksn4397, peter8526, yunseyoung}@kaist.ac.kr
https://flex-judge.github.io

## Abstract

Human-generated reward signals are critical for aligning generative models with human preferences, guiding both training and inference-time evaluations. While large language models (LLMs) employed as proxy evaluators, *i.e.,* LLM-as-a-Judge, significantly reduce the costs associated with manual annotations, they typically require extensive modality-specific training data and fail to generalize well across diverse multimodal tasks. In this paper, we propose **FLEX-Judge**, a reasoning-guided multimodal judge model that leverages minimal textual reasoning data to robustly generalize across multiple modalities and evaluation formats. Our core intuition is that structured textual reasoning explanations inherently encode generalizable decision-making patterns, enabling an effective transfer to multimodal judgments, *e.g.,* with images or videos. Empirical results demonstrate that FLEX-Judge, despite being trained on significantly fewer text data, achieves competitive or superior performance compared to state-of-the-art commercial APIs and extensively trained multimodal evaluators. Notably, FLEX-Judge presents broad impact in modalities like molecule, where comprehensive evaluation benchmarks are scarce, underscoring its practical value in resource-constrained domains. Our framework highlights reasoning-based text supervision as a powerful, cost-effective alternative to traditional annotation-intensive approaches, substantially advancing scalable multimodal model-as-a-judge.

## 1   Introduction

Human-generated reward signals play a crucial role in both training and deploying generative models. They are commonly used to fine-tune models toward human-aligned behavior through preference optimization [48, 56] or reinforcement learning [61, 91]. At inference time, they also guide inference-time decisions, *e.g.,* best-of-$N$ selection [24], output reranking [43], or filtering based on quality or safety criteria, making them essential tools for test-time control. As models become more capable and are applied across diverse modalities and tasks, the need for high-quality, consistent human feedback continues to grow. However, scaling human feedback process is highly resource-intensive and challenging to generalize across different domains, highlighting a critical demand for more scalable and cost-effective alternatives that can reliably approximate human judgments [71, 89].

A promising alternative to manual feedback collection is to use large language models (LLMs) as proxy evaluators—an approach known as LLM-as-a-Judge [89]. These models emulate human preferences via instruction-following prompts and have shown strong agreement with human ratings across tasks such as summarization [60, 61] and dialogue [29]. In addition to reducing annotation costs, they can serve as reusable, modular evaluators and are often comparable to human judges in consistency. However, existing approaches are largely restricted to text-only scenarios [36] and often

---

[*]Two authors are equally contributed

39th Conference on Neural Information Processing Systems (NeurIPS 2025).

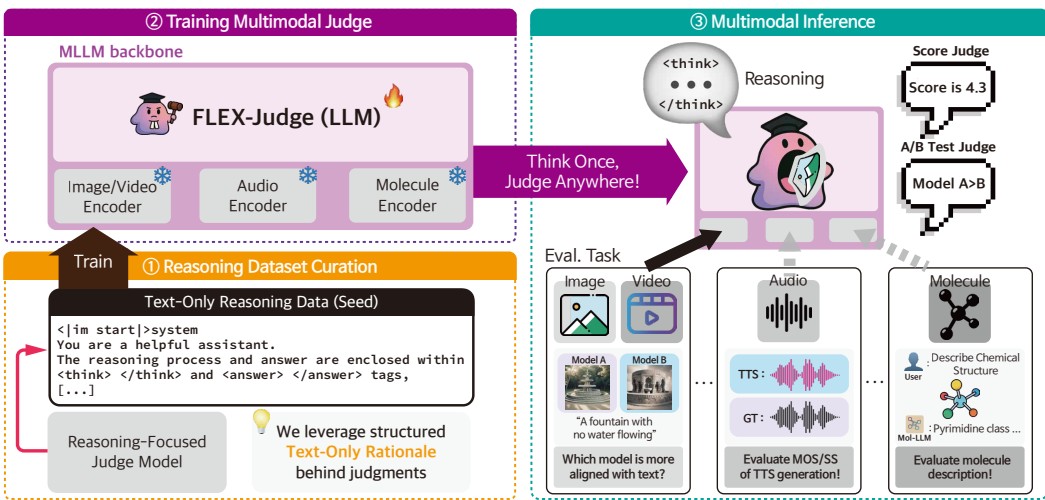

Figure 1: Conceptual overview of FLEX-Judge. We train a multimodal judge model using a small amount of text-only reasoning data. Unlike previous approaches that require modality-specific supervision, **FLEX-Judge leverages structured text-only rationale behind judgments to enable generalization across modalities.** Once trained, FLEX-Judge can be applied to various evaluation tasks, including vision-language tasks, audio quality scoring, and molecular structure, without the need for additional task-specific or modality-specific annotations.

require substantial amounts of paired preference data [29] to generalize across evaluation types, such as single-score grading, pairwise comparison, or batch-level assessments.

Extending this paradigm to multimodal domains (*e.g.,* vision-language generation or image caption ranking) presents unique challenges. Although some efforts have adapted LLM-as-a-Judge models into vision-language evaluators [11], they typically require extensive modality-specific annotations [76] and frequently fail to generalize across diverse formats without re-training or fine-tuning on each new task [32]. Moreover, the lack of publicly available multimodal preference datasets makes it difficult to systematically evaluate and train such models, often resulting in heavy reliance on proprietary models or benchmarks.

Motivated by the these challenges, we ask a key question: *Can just a small amount of textual reasoning data serve to train a cost-efficient, modality-agnostic judge model?* Our core intuition is that textual reasoning chains, such as evaluative explanations or comparisons behind preference judgments, encode structured and interpretable rationale that can be transferred across modalities and evaluation formats. That is, models trained to make judgments by reasoning about why one answer is preferred over another may learn more generalizable decision rules. In addition, recent advances in multimodal large language models (MLLMs) suggest that their impressive generalization capabilities predominantly arise from their pretrained textual reasoning abilities.

**Contribution & Organization.** Inspired by this, we propose **FLEX-Judge**, a multimodal judge model trained solely on a small corpus of high-quality text reasoning data. Our central finding is that: *we do not need large-scale multimodal annotations to train an effective MLLM judge—just a small amount of good reasoning data is enough.* This text-only supervision is not only cheaper but also avoids the need for complex annotation tools or multimodal data curation, while achieving strong generalization across diverse modalities and evaluation settings. Refer to Figure 1 for the preview of this work. Our key contributions are:

- **Modality-agnostic Efficient Approach (Section 2):** We propose FLEX-Judge, a simple and cost-effective method that uses 1K-sized text-only reasoning data to generalize across modalities without modality-specific training. This facilitates zero-shot evaluation on unseen modalities with minimal annotation and compute overhead.

- **Comparison with State-of-the-art (Section 3):** We evaluate FLEX-Judge on image, video, and audio reward benchmarks against commercial APIs [1, 62], large vanilla MLLMs, and open-source judges trained on costly multimodal datasets [32, 76]. Despite its simplicity and efficiency, FLEX-Judge (7B model) outperforms open-source judges, even exceeding Gemini and GPT-4o on several MJ-Bench and GenAI-Bench subtasks. In-depth analyses are presented in Section 5.

- **Broader Impact (Section 4):** We demonstrate real-world applications of FLEX-Judge in the molecular domain by introducing FLEX-Mol-LLaMA, the first judge model designed for molecular modality [27]. We showcase its utility in two key scenarios: (1) serving as a best-of-N selector for inference-time scaling, and (2) constructing training data for direct preference optimization (DPO) [56]. In both cases, reward-guided molecular MLLM achieves significant improvements, highlighting the practical solution in domains where modality-specific reward models are infeasible.

## 2 Approach

### 2.1 Motivation

**Problem Statement.** Evaluating outputs across multiple modalities using foundation models is increasingly important, especially as generative models expand beyond language to include image, video, or audio [4, 16, 79]. While both proprietary LMs and open-source evaluators are widely used to assess the generative models' response quality, two main challenges remain:

- **Concern of Commercial API:** Issues regarding transparency, controllability, and affordability persist when utilizing proprietary LMs for evaluation tasks [29]. API model changes can silently degrade evaluation quality, raising concerns about the reliability of LLM-as-a-Judge with closed-source models [9]. For instance, Xiong et al. [76] re-evaluated GPT-4V on the MLLM-as-a-Judge benchmark [11] and found a significant drop in evaluation performance.
- **Limited Support for Diverse Modalities:** While judge models for language [29] and vision-language tasks [32, 76] have advanced significantly thanks to the availability of assessment data, evaluating other modalities, *e.g.,* audio [15, 16], thermal heatmaps [84], 3D point clouds [25], and molecular structures [41], remains underexplored, with few effective judge models or publicly available training resources. For instance, evaluating a state-of-the-art molecular LLM [27] often relies on GPT-4o to assess the soundness and relevance of its responses, which is not only hard to reproduce but also unreliable, as GPT-4o cannot handle molecular modalities.

**Hypothesis.** In multilingual LLMs [20, 53, 74], it has been observed that fine-tuning on downstream tasks in one language can lead to performance improvements in other languages as well, demonstrating cross-lingual generalization. This suggests that when a shared representation space exists, task knowledge can effectively transfer across different languages.

We hypothesize that a similar phenomenon may occur in multimodal settings: *if a model learns a unified cross-modal representation, then fine-tuning on a single modality—especially text—may enable generalization to other modalities.* However, such investigations on cross-modal transfer are still rare in MLLMs. Motivated by findings from multilingual LLMs, we propose to build a practical multimodal judge model using a small amount of text-based reasoning data and demonstrate that this model can be applied across a variety of modalities, including those with scarce data resources.

### 2.2 FLEX-Judge: Reasoning-Guided MLLM as a Judge

Building on our hypothesis, we propose **FLEX-Judge**, a multimodal judge model framework trained *a priori* through textual reasoning annotations. Unlike existing judge models that heavily depend on extensive modality-specific preference data, our framework strategically leverages a small but carefully curated corpus of *reasoning* data (THINK ONCE)—explanatory textual annotations indicating why certain outputs are preferred over others—to foster robust evaluation capabilities across diverse modalities (JUDGE ANYWHERE).

**Data Curation.** We begin by generating a high-quality "seed" textual dataset leveraging JudgeLRM [12], a reasoning-focused judge LM explicitly trained to evaluate AI responses with structured explanations. JudgeLRM uses crafted prompt templates to assess single or paired AI-generated outputs, producing detailed rationales enclosed within specialized tags (`<think></think>`). These reasoning annotations are detailed and comprehensive, explicitly addressing criteria such as correctness, completeness, consistency, relevance, and coherence.

A critical advantage of our framework is the minimal data requirement. Specifically, we rely on only a *1K-sized* corpus of high-quality textual reasoning annotations on text-only evaluation samples, making our approach highly cost-efficient, compared to MLLM judges such as Prometheus-Vision [32] (150K image-text evaluation pairs for training) and LLaVA-Critic [76] (113K pairs). This corpus is carefully curated to exhibit the following properties:

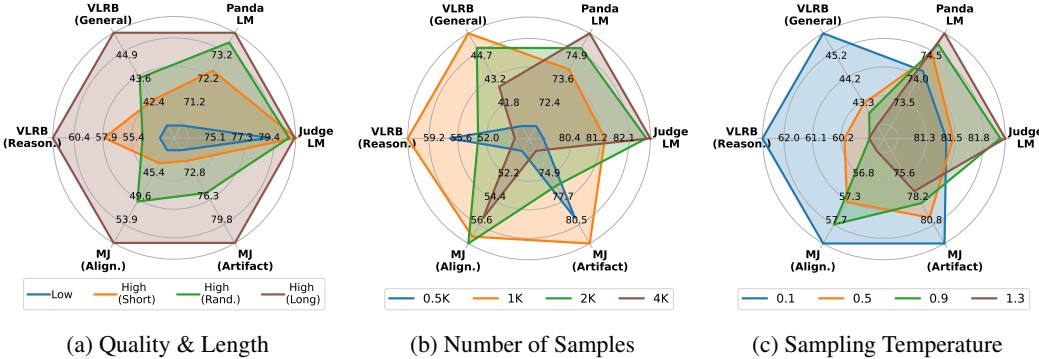

| (a) Quality & Length | (b) Number of Samples | (c) Sampling Temperature |

Figure 2: Comparisons on different perspectives of the seed dataset curation. All evaluations are done with our FLEX-Judge, where its backbone model is Qwen2.5-VL-7B (refer to Section 3.1 for model details) and has been trained on the JudgeLRM-7B response data.

- **Quality & Difficulty:** Following Muennighoff et al. [50], we prioritize high-quality and high-difficulty samples when curating a training dataset for better sample efficiency. Specifically, we utilize JudgeLRM-7B to generate evaluation responses for prompts in the `JudgeLM-100K` dataset [90], and filter out responses whose predicted ratings mismatch with those annotated by GPT-4o in the original dataset. This quality-based selection process significantly improves the performance of the trained judge compared to random selection (*i.e.,* blue; low-quality in Figure 2a). Consistent with the findings of [50], we also observe that samples with longer reasoning chains (*i.e.,* brown) improve judge performance across both language-only and multimodal settings. This observation supports the hypothesis that longer reasoning indicates higher difficulty, thereby enhancing the effectiveness of limited training data.

- **Number of Data Samples & On-policy:** We observe that training with a large number of samples can cause *catastrophic forgetting* [81], where the LLM backbone exhibits diminished capacity to encode visual or audio features. As shown in Figure 2b, while JudgeLM [90] and PandaLM [70] performances improve with more training data, performances on multimodal benchmarks (MLLM-as-a-Judge and MJ-Bench) degrade, indicating a modality shift detrimental to multimodal understanding. We also find in Figure 2c that lower-temperature decoding yields more effective training data, with lower initial losses. Since JudgeLRM-7B (the data generator) shares its LLM backbone with FLEX-Judge, using these lower-loss, on-policy samples helps prevent catastrophic forgetting while preserving the language-side judge performance.

- **Format Diversity:** Compared to naïvely using unprocessed outputs from JudgeLRM-7B, which only supports pairwise scoring on a 1–10 scale as shown in Figure 7, we post-process the model's outputs to support both single-score and pairwise grading, with scores mapped to either 1–10 or 1–5 scales depending on the instruction. FLEX-Judge trained on these post-processed outputs demonstrates improved generalization to diverse evaluation formats, including a single-score grading and a batch-level ranking (*i.e.,* where the number of response options exceeds three) as used in [11]. Furthermore, we find that the post-processed variant is more robust to varied instruction styles, particularly when prompts emphasize different evaluation criteria across input pairs. The detailed results are provided in Section 3.2 (Table 1).

**Training Multimodal Judge.** Next, we use the reasoning seed dataset to fine-tune an MLLM, such as Qwen2.5-VL [5] and Qwen2.5-Omni [79]. Despite being originally trained to handle both text and other modalities such as visual or audio inputs, the MLLM is fine-tuned exclusively on our textual reasoning annotations. We call this fine-tuned evaluator as FLEX-Judge. Consequently, the structured and explicit reasoning provided by JudgeLRM enables FLEX-Judge to learn how to systematically evaluate and justify preferences, significantly improving zero-shot transfer capabilities.

**Multimodal Inference: Reasoning-Guided Preference Judgments.** At inference time, our model performs multimodal evaluation without additional fine-tuning or modality-specific annotations (*i.e.,* training-free). We visualize an example output from FLEX-Judge in Figure 3, demonstrating that the textual reasoning behind judgments has transferred across modalities. Furthermore, unlike most other evaluators, our FLEX-Judge takes advantage of *inference-time scaling* to improve judgment performance by leveraging multiple reasoning paths, such as majority voting [69] or budget forcing [50], as addressed in Section 5.

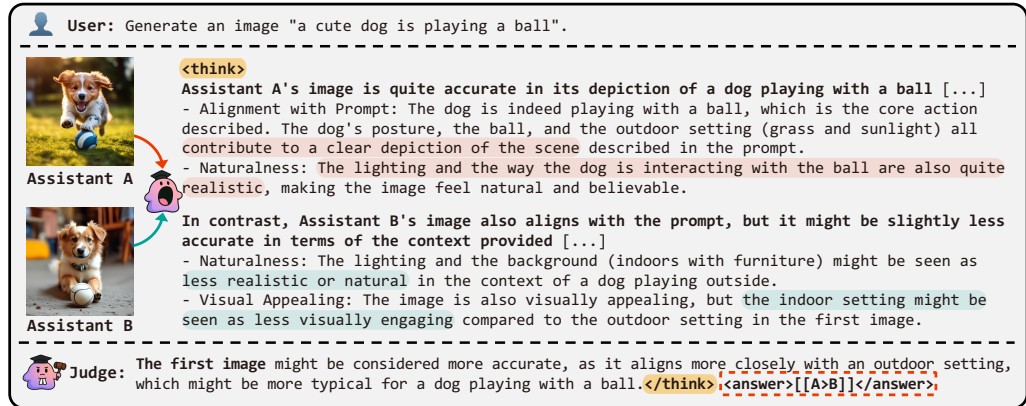

Figure 3: Reasoning process of FLEX-Judge on the text-image alignment task (GenAI-Bench [35]). Additional qualitative examples are found in Appendix D.

## 3 Experimental Evaluation of FLEX-Judge

We comprehensively evaluate FLEX-Judge across diverse modalities, including images, videos, and audio, demonstrating its generalization capability and competitive performance against state-of-the-art judge models. Notably, it matches closed-source commercial APIs on vision tasks and outperforms all training-free evaluators in audio understanding. These strong results suggest that FLEX-Judge can be used with confidence in modalities even when expert judge models are not applicable, which we further explore in Section 4.

### 3.1 Experimental Setup

**Implementation.** Based on *1K-sized* training dataset introduced in Section 2.2, we develop FLEX-Omni-7B (image, video, and audio) and FLEX-VL-7B (image and video) from Qwen2.5-Omni-7B [79] and Qwen2.5-VL-7B [5], respectively. We compare them against both commercial models with costly API usage [1, 62] and open-source models that require either extensive training data [32, 76] or significantly more parameters [40, 67]. For more implementation details, refer to Appendix B.

**Evaluation Protocol for Judge Models.** To evaluate the quality of judge models, we measure how closely their assessments align with human annotations (or human-verified model evaluations [14, 38]). The specific metric depends on the evaluation format: we measure (1) Pearson correlation [33] for single-score grading tasks, (2) accuracy for pairwise (A/B) comparisons, and (3) normalized Levenshtein distance [34] for batch-level rankings (*e.g.,* ABCD). Detailed descriptions for each benchmark and the evaluation prompts are provided in Appendix B.3.

**Evaluation Benchmarks.** We evaluate image understanding capabilities of FLEX-Judge using the MLLM-as-a-Judge benchmark [11], which comprises 14 diverse vision-language tasks including captioning and website browsing, and the VL-RewardBench benchmark[1] [38], which focuses on complex reasoning tasks like visual hallucination detection. For image generation assessment, we use MJ-Bench [14] to assess image quality and alignment. We use GenAI-Bench [35] for evaluating video generation and image editing. For audio understanding, following the prior work [68], we conduct speech quality assessment task, specifically performing mean opinion score (MOS) prediction by using the NISQA [49], BVCC [18], and SOMOS [46] datasets, and speaker similarity score (SS) prediction with VoxSim [2] dataset for assessing speaker similarity score (SS). For additional results of Multimodal RewardBench [82] and JudgeAnything [54] benchmarks, refer to Appendix C.1. We also provide the language-only assessment results in Appendix C.4.

### 3.2 Comparison with State-of-the-arts

**Image Understanding.** In Table 1, model judgments are compared with human ratings in MLLM-as-a-Judge benchmark. We compare FLEX-Judge with the following model groups: commercial

---

[1]Disclaimer: We completed all experiments on May 9th, but both VL-RewardBench and MJ-Bench were later modified on May 13th and 16th, respectively.

Table 1: The overall performance of different MLLMs in judging, compared with human annotations on different datasets. We sample all judgments three times and average them to mitigate the bias. w. and w.o. tie represents tie and non-tie situations, respectively, by following [11]. ◇: reported results from LLaVA-Critic [76]. ♠: results from the original paper of MLLM-as-a-Judge [11]. †: Prometheus-Vision-13B [32] is only trained under the Score setting, incapable of following Pair/Batch instructions. *Training-free (TF)* models have not been trained on multimodal evaluation data.

| | Model | TF? | COCO | C.C. | Diff. | Graphics | Math | Text | WIT | Chart | VisIT | CC-3M | M2W | SciQA | Aes | MM-Vet | Ave. |
|---|---|---|---|---|---|---|---|---|---|---|---|---|---|---|---|---|---|
| **Score (↑)** | GPT-4V◇ | - | 0.410 | 0.444 | 0.361 | 0.449 | 0.486 | 0.506 | 0.457 | 0.585 | 0.554 | 0.266 | 0.267 | 0.315 | 0.472 | 0.367 | 0.424 |
| | Gemini-1.0-Pro-Vision♠ | - | 0.262 | 0.408 | - | 0.400 | 0.228 | 0.222 | 0.418 | 0.343 | 0.336 | 0.374 | 0.324 | 0.073 | 0.360 | 0.207 | 0.304 |
| | Gemini-2.5-Pro | - | 0.409 | 0.426 | 0.467 | 0.559 | 0.471 | 0.553 | 0.254 | 0.636 | 0.563 | 0.254 | 0.073 | 0.600 | 0.139 | 0.058 | 0.390 |
| | Prometheus-V-13B◇† | ✗ | 0.289 | 0.342 | 0.106 | 0.172 | 0.182 | 0.214 | **0.209** | 0.224 | 0.226 | 0.228 | 0.089 | 0.174 | 0.368 | 0.157 | 0.213 |
| | LLaVA-Critic-7B◇ | ✗ | **0.382** | **0.450** | 0.103 | 0.316 | 0.356 | 0.378 | 0.179 | 0.421 | 0.322 | 0.246 | 0.301 | 0.269 | 0.395 | 0.272 | 0.314 |
| | LLaVA-1.6-34B♠ | ✓ | 0.285 | 0.251 | -0.012 | 0.262 | 0.238 | 0.258 | 0.151 | 0.318 | 0.198 | 0.109 | 0.022 | 0.206 | 0.025 | 0.265 | 0.184 |
| | Qwen2.5-Omni-7B | ✓ | 0.150 | 0.017 | -0.045 | 0.087 | -0.003 | 0.049 | 0.060 | -0.010 | 0.136 | 0.073 | 0.136 | 0.097 | 0.148 | 0.108 | 0.072 |
| | Qwen2.5-VL-7B | ✓ | 0.294 | 0.247 | -0.020 | -0.041 | 0.095 | 0.170 | 0.056 | 0.011 | 0.328 | 0.178 | 0.255 | 0.311 | 0.327 | 0.103 | 0.165 |
| | FLEX-Omni-7B | ✓ | 0.324 | 0.281 | **0.126** | **0.371** | 0.116 | **0.429** | 0.118 | **0.501** | **0.479** | **0.275** | **0.375** | 0.351 | 0.309 | 0.232 | 0.306 |
| | FLEX-VL-7B | ✓ | 0.363 | 0.235 | 0.114 | 0.338 | **0.448** | 0.423 | 0.125 | 0.471 | 0.452 | 0.189 | 0.357 | **0.380** | **0.407** | **0.343** | **0.332** |
| **Pair w. Tie (↑)** | GPT-4V◇ | - | 0.539 | 0.634 | 0.668 | 0.632 | 0.459 | 0.495 | 0.536 | 0.369 | 0.591 | 0.544 | 0.544 | 0.389 | 0.620 | 0.517 | 0.538 |
| | Gemini-1.0-Pro-Vision♠ | - | 0.616 | 0.787 | - | 0.650 | 0.436 | 0.664 | 0.605 | 0.500 | 0.660 | 0.560 | 0.370 | 0.262 | 0.190 | 0.312 | 0.509 |
| | Gemini-2.5-Pro | - | 0.540 | 0.606 | 0.753 | 0.618 | 0.455 | 0.532 | 0.508 | 0.370 | 0.604 | 0.555 | 0.660 | 0.365 | 0.690 | 0.527 | 0.556 |
| | LLaVA-Critic-7B◇ | ✗ | **0.593** | **0.687** | 0.707 | **0.587** | 0.432 | 0.544 | **0.564** | 0.338 | 0.596 | 0.628 | 0.591 | 0.370 | **0.686** | 0.464 | **0.556** |
| | LLaVA-1.6-34B♠ | ✓ | 0.493 | 0.600 | 0.570 | 0.300 | 0.374 | **0.551** | 0.543 | 0.254 | 0.398 | 0.392 | 0.513 | 0.434 | 0.524 | 0.499 | 0.460 |
| | Qwen2.5-Omni-7B | ✓ | 0.462 | 0.479 | **0.733** | 0.422 | 0.385 | 0.432 | 0.411 | 0.394 | 0.489 | 0.501 | 0.508 | 0.395 | 0.517 | 0.462 | 0.471 |
| | Qwen2.5-VL-7B | ✓ | 0.446 | 0.474 | 0.507 | 0.326 | 0.397 | 0.383 | 0.366 | 0.364 | 0.461 | 0.483 | 0.358 | **0.442** | 0.494 | 0.420 | 0.423 |
| | FLEX-Omni-7B | ✓ | 0.496 | 0.647 | 0.713 | 0.490 | 0.429 | 0.485 | 0.445 | **0.432** | 0.592 | 0.579 | 0.593 | 0.384 | 0.636 | **0.524** | 0.532 |
| | FLEX-VL-7B | ✓ | 0.538 | 0.685 | 0.653 | 0.532 | **0.446** | 0.534 | 0.458 | 0.386 | 0.586 | 0.586 | **0.595** | 0.391 | 0.636 | 0.500 | 0.538 |
| **Pair w.o. Tie (↑)** | GPT-4V◇ | - | 0.729 | 0.772 | 0.884 | 0.853 | 0.665 | 0.661 | 0.760 | 0.495 | 0.785 | 0.707 | 0.697 | 0.639 | 0.741 | 0.654 | 0.717 |
| | Gemini-1.0-Pro-Vision♠ | - | 0.717 | 0.840 | - | 0.770 | 0.678 | 0.793 | 0.688 | 0.658 | 0.711 | 0.652 | 0.471 | 0.358 | 0.265 | 0.400 | 0.615 |
| | Gemini-2.5-Pro | - | 0.699 | 0.677 | 0.783 | 0.768 | 0.518 | 0.611 | 0.604 | 0.513 | 0.724 | 0.649 | 0.740 | 0.645 | 0.709 | 0.705 | 0.668 |
| | LLaVA-Critic-7B◇ | ✗ | **0.771** | 0.774 | 0.755 | **0.758** | **0.596** | 0.658 | 0.680 | 0.488 | **0.727** | **0.742** | 0.692 | 0.658 | **0.715** | 0.635 | **0.689** |
| | LLaVA-1.6-34B♠ | ✓ | 0.607 | **0.824** | **0.855** | 0.402 | 0.587 | **0.750** | **0.758** | 0.381 | 0.503 | 0.564 | **0.712** | 0.679 | 0.694 | **0.762** | 0.648 |
| | Qwen2.5-Omni-7B | ✓ | 0.559 | 0.518 | 0.755 | 0.478 | 0.407 | 0.456 | 0.464 | 0.443 | 0.557 | 0.557 | 0.545 | 0.617 | 0.525 | 0.489 | 0.526 |
| | Qwen2.5-VL-7B | ✓ | 0.479 | 0.492 | 0.510 | 0.268 | 0.394 | 0.394 | 0.334 | 0.348 | 0.506 | 0.538 | 0.330 | 0.511 | 0.486 | 0.388 | 0.425 |
| | FLEX-Omni-7B | ✓ | 0.648 | 0.720 | 0.748 | 0.621 | 0.560 | 0.566 | 0.534 | **0.609** | 0.711 | 0.679 | 0.666 | **0.706** | 0.655 | 0.674 | 0.650 |
| | FLEX-VL-7B | ✓ | 0.689 | 0.763 | 0.685 | 0.670 | 0.580 | 0.613 | 0.542 | 0.548 | 0.702 | 0.686 | 0.666 | 0.684 | 0.655 | 0.683 | 0.655 |
| **Batch (↓)** | GPT-4V♠ | - | 0.318 | 0.353 | 0.070 | 0.385 | 0.348 | 0.319 | 0.290 | 0.347 | 0.300 | 0.402 | 0.597 | 0.462 | 0.453 | 0.411 | 0.361 |
| | Gemini-1.0-Pro-Vision♠ | - | 0.287 | 0.299 | - | 0.473 | 0.462 | 0.430 | 0.344 | 0.520 | 0.426 | 0.357 | 0.613 | 0.412 | 0.467 | 0.529 | 0.432 |
| | Gemini-2.5-Pro | - | 0.517 | 0.509 | 0.290 | 0.595 | 0.599 | 0.532 | 0.488 | 0.557 | 0.530 | 0.505 | 0.532 | 0.501 | 0.503 | 0.512 | 0.512 |
| | LLaVA-Critic-7B | ✗ | 0.541 | 0.455 | 0.525 | 0.612 | 0.576 | 0.599 | 0.603 | 0.580 | 0.481 | 0.592 | 0.588 | 0.627 | 0.618 | 0.515 | 0.565 |
| | LLaVA-1.6-34B♠ | ✓ | 0.449 | 0.411 | 0.500 | 0.561 | 0.575 | 0.544 | 0.483 | 0.552 | 0.542 | 0.479 | 0.529 | **0.437** | 0.500 | 0.450 | 0.501 |
| | Qwen2.5-Omni-7B | ✓ | 0.545 | 0.518 | 0.635 | 0.591 | 0.589 | 0.602 | 0.588 | 0.545 | 0.582 | 0.538 | 0.594 | 0.576 | 0.574 | 0.581 | 0.576 |
| | Qwen2.5-VL-7B | ✓ | 0.562 | 0.450 | 0.593 | 0.630 | 0.607 | 0.582 | 0.631 | 0.570 | 0.569 | 0.519 | 0.639 | 0.703 | 0.558 | 0.572 | 0.585 |
| | FLEX-Omni-7B | ✓ | **0.392** | 0.328 | **0.404** | 0.452 | 0.439 | 0.417 | **0.462** | 0.455 | **0.342** | 0.450 | 0.442 | 0.484 | 0.515 | **0.362** | **0.425** |
| | FLEX-VL-7B | ✓ | 0.419 | **0.325** | 0.414 | 0.462 | **0.437** | 0.412 | 0.477 | **0.445** | 0.398 | **0.392** | 0.420 | 0.487 | **0.471** | 0.405 | 0.426 |

Table 2: Comparison of MLLM evaluator performances on VL-RewardBench (*Left*) and MJ-Bench (*Right*). ◇: results from the original works [14, 38]. **Best** and second best results.

| Model | TF? | General | Hallu. | Reason. | Overall | Macro |
|---|---|---|---|---|---|---|
| GPT-4o◇ | - | 49.1 | 67.6 | 70.5 | 65.8 | 62.4 |
| Gemini-1.5-Pro◇ | - | 50.8 | 72.5 | 64.2 | 62.5 | 58.4 |
| Gemini-2.5-Pro | - | 44.3 | 49.1 | 53.0 | 48.4 | 48.8 |
| LLaVA-OneVision-7B◇ | ✓ | 32.2 | 20.1 | 57.1 | 29.6 | 36.5 |
| InternVL2-8B◇ | ✓ | 35.6 | 41.1 | 59.0 | 44.5 | 45.2 |
| Qwen2.5-Omni-7B | ✓ | 32.6 | 18.3 | 28.7 | 23.5 | 26.6 |
| Qwen2.5-VL-7B | ✓ | 37.7 | 33.1 | 48.2 | 36.3 | 39.7 |
| Pixtral-12B◇ | ✓ | 35.6 | 25.9 | 59.9 | 35.8 | 40.4 |
| Qwen2-VL-72B◇ | ✓ | 38.1 | 32.0 | 61.0 | 39.5 | 43.0 |
| Molmo-72B◇ | ✓ | 38.3 | 42.5 | 62.6 | 44.1 | 43.7 |
| NVLM-D-72B◇ | ✓ | 38.9 | 31.6 | 62.0 | 40.1 | 44.3 |
| LLaVA-Critic-7B | ✗ | **47.4** | 38.5 | 53.8 | 43.7 | 46.6 |
| FLEX-Omni-7B | ✓ | 47.01 | 42.72 | 61.08 | 48.02 | 50.27 |
| FLEX-VL-7B | ✓ | 46.11 | **43.39** | **62.87** | **48.60** | **50.79** |

| Model | TF? | Alignment | | Safety | | Artifact | |
|---|---|---|---|---|---|---|---|
| | | w. Tie | w.o. Tie | w. Tie | w.o. Tie | w. Tie | w.o. Tie |
| GPT-4o◇ | - | 61.5 | 62.5 | 35.3 | 100.0 | 97.6 | 98.7 |
| Gemini Ultra◇ | - | 67.2 | 69.0 | 13.1 | 95.1 | 55.7 | 96.7 |
| Claude 3 Opus◇ | - | 57.1 | 55.9 | 13.4 | 78.9 | 11.9 | 70.4 |
| PickScore-v1◇ | ✗ | 58.8 | 64.6 | 37.2 | 42.2 | **83.8** | 89.6 |
| HPS-v2.1◇ | ✗ | 47.3 | **70.1** | 18.8 | 41.3 | 67.3 | **93.5** |
| ImageReward◇ | ✗ | 50.9 | 64.7 | 24.9 | 38.7 | 63.5 | 81.8 |
| LLaVA-1.6-13B◇ | ✗ | 29.1 | 60.3 | 27.9 | 45.6 | 36.8 | 62.5 |
| Prometheus-Vision-13B◇ | ✗ | 11.8 | 64.3 | 28.6 | **71.4** | 8.7 | 67.9 |
| FLEX-Omni-7B | ✓ | **60.84** | 62.46 | 47.69 | 65.21 | 75.80 | 91.66 |
| FLEX-VL-7B | ✓ | 58.16 | 59.13 | **57.51** | 66.88 | 82.32 | 89.08 |

models including Gemini and GPT-4V (state-of-the-art closed-source models), and latest open-source baselines including Prometheus-Vision-7B [32] and LLaVA-Critic-7B [76], trained on large-scale (>100K) curated MLLM-as-a-judge datasets.

We highlight that prior open-source judges are limited to specific evaluation formats, *e.g.,* LLaVA-Critic-7B faltering in batch-level ranking, making them less utilitarian. In contrast, our models are capable of handling diverse evaluation criteria, matching or outperforming much larger models like Gemini, GPT-4V, and LLaVA-1.6-34B. Notably, LLaVA-Critic-7B, trained on 113K vision-language understanding data, underperforms both FLEX-Omni-7B and FLEX-VL-7B on VL-RewardBench (see Table 2; *left*). These results mark the simplicity and efficiency of our approach, learning from 1K text reasoning annotations without any modality-specific supervision (training-free; TF).

**Image Generation.** Table 2 (*right*) presents the judge performance of generated images in MJ-Bench, whether they are well-aligned with a prompt, safe, or have artifacts. FLEX-Judge models achieve higher scores than some commercial models (*e.g.,* Claude 3 Opus) and all training-required judge models (PickScore-v1 [30], HPS-v2.1 [75], and ImageReward [78]) by a large margin. While the baselines have high variance according to tasks, especially poor in safety check, our models perform highly consistent. The superiority of FLEX-Judge in image generation assessment is further demonstrated in Table 3, where FLEX-VL-7B with majority voting evaluation [69] outperforms GPT-4o and Gemini-1.5-Pro.

**Image Editing & Video Generation.** We further report judge performance on GenAI-Bench, including image generation, image editing, and video generation tasks, in Table 3. Notably, our FLEX-VL-7B achieves the highest overall performance, as well as strong results in both image editing and video generation tasks, when inference-time scaling is applied [69]. A detailed discussion of inference-time scaling is provided in Section 5. While FLEX-Omni-7B shows lower performance initially, it also benefits significantly from inference-time scaling—unlike its non-reasoning baseline, Qwen2.5-VL-7B.

Table 3: Comparison of MLLM evaluator performance on GenAI-Bench. $\diamond$: results from the original work [35].

| Model | Image Gen. | Image Edit. | Video Gen. | Overall |
|---|---|---|---|---|
| GPT-4o$\diamond$ | 45.59 | 53.54 | 48.46 | 49.20 |
| Gemini-1.5-Pro$\diamond$ | 44.67 | 55.93 | 46.21 | 48.94 |
| Gemini-2.5-Pro | **47.55** | **65.51** | **50.33** | **54.46** |
| LLaVA$\diamond$ | 37.00 | 26.12 | 30.40 | 31.17 |
| LLaVA-NeXT$\diamond$ | 22.65 | 25.35 | 21.70 | 23.23 |
| Qwen2.5-Omni-7B | 34.87 | 31.88 | 38.45 | 35.07 |
| Qwen2.5-VL-7B | 31.93 | 38.63 | 37.61 | 36.06 |
| FLEX-Omni-7B | 38.15 | 46.73 | 37.10 | 40.66 |
| + Majority Voting [69] | 41.67 | 52.01 | 44.25 | 45.98 |
| FLEX-VL-7B | 43.32 | 47.41 | 44.78 | 45.17 |
| + Majority Voting [69] | 46.34 | 54.19 | 47.34 | 49.29 |

**Audio Understanding.** Table 4 demonstrates the performance of our approach in speech quality evaluation, where the linear correlation coefficient (LCC) and Spearman's rank correlation coefficient (SRCC) are calculated to assess the agreement between the model's predicted scores and the human-annotated MOS and SS values. As observed in [10, 68, 86], existing open-source audio LLMs like Qwen2-Audio struggle in quality assessment without specific fine-tuning, often exhibiting widespread hallucinations. While audio LLMs are primarily pretrained on semantics-related tasks like audio question answering (AQA) or captioning (AAC), they are less familiar with such qualitative evaluations, which are challenging due to different MOS standards depending on the dataset [68]. Still, FLEX-Omni-7B outperforms all training-free judges and even Gemini-2.0-Flash.

Table 4: Audio MOS/SS prediction results on the test sets of the NISQA [49], BVCC [18], SO-MOS [46], and VoxSim [2] datasets. System-level results are computed by averaging the utterance-level results within each text-to-speech system (not provided for NISQA due to the absence of system labels). $\diamond$: task-specific fine-tuning results from [68].

| Model | TF? | NISQA (MOS) utterance-level | | BVCC (MOS) utterance-level | | BVCC (MOS) system-level | | SOMOS (MOS) utterance-level | | SOMOS (MOS) system-level | | VoxSim (SS) utterance-level | |
|---|---|---|---|---|---|---|---|---|---|---|---|---|---|
| | | LCC | SRCC | LCC | SRCC | LCC | SRCC | LCC | SRCC | LCC | SRCC | LCC | SRCC |
| Gemini-2.0-Flash | - | 0.408 | 0.415 | 0.044 | 0.038 | 0.092 | 0.096 | 0.119 | 0.129 | 0.256 | 0.317 | 0.451 | 0.457 |
| Gemini-2.5-Pro | - | 0.567 | 0.586 | 0.261 | 0.266 | 0.495 | 0.504 | 0.193 | 0.186 | 0.434 | 0.436 | 0.661 | 0.667 |
| Single-task SOTA$\diamond$ | ✗ | 0.894 | 0.887 | 0.899 | 0.896 | 0.939 | 0.936 | 0.687 | 0.681 | 0.911 | 0.917 | 0.835 | 0.836 |
| Qwen2-Audio$\diamond$ | ✗ | 0.768 | 0.780 | 0.681 | 0.678 | 0.800 | 0.797 | 0.583 | 0.572 | 0.850 | 0.873 | 0.415 | 0.505 |
| Qwen2.5-Omni-7B | ✓ | 0.210 | 0.243 | 0.056 | 0.055 | -0.075 | -0.079 | 0.074 | 0.097 | 0.136 | 0.159 | 0.211 | 0.204 |
| Qwen2-Audio | ✓ | 0.004 | -0.002 | -0.067 | -0.062 | 0.093 | 0.120 | -0.016 | -0.013 | 0.058 | 0.077 | 0.042 | 0.044 |
| FLEX-Omni-7B | ✓ | **0.545** | **0.590** | **0.081** | **0.067** | **0.144** | **0.128** | **0.150** | **0.138** | **0.323** | **0.325** | **0.271** | **0.268** |

## 4 Broader Impact: Case Study on Molecule Evaluator

Trained only on textual reasoning data without any modality-specific supervision, FLEX-Judge demonstrates strong generalization across image, video, and audio tasks. This suggests that FLEX-Judge can serve as a practical solution in domains where constructing modality-specific reasoning datasets is infeasible or prohibitively expensive.

A particularly compelling use case arises in scientific domains such as molecular modeling [3, 58, 83], where no existing reward model or judge is available due to limited data and domain complexity. To test our hypothesis, we build a molecular judge using Mol-LLaMA [27], which is based on LLaMA3.1-8B [23] and understands both 2D and 3D molecule representations. We fine-tune its LLM backbone as a reasoning-aware judge, while keeping its modality-aware modules (LoRA

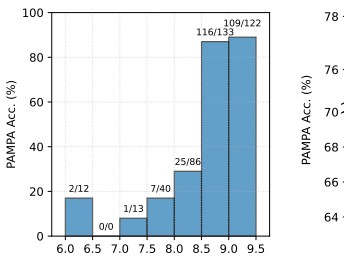 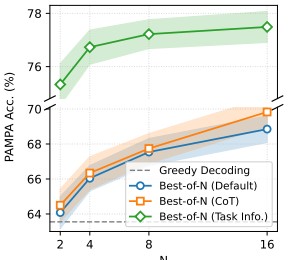

| Model | Default | CoT | w/ Task Info. |
|---|---|---|---|
| Llama3.1-8B-Instruct | 56.51 | 46.19 | 63.64 |
| Mol-Instructions [22] | 55.91 | 33.50 | 70.47 |
| 3D-MoLM [39] | 46.93 | 50.00 | 64.86 |
| LLaMo [52] | 49.25 | 64.37 | 48.51 |
| Mol-LLaMA [27] | 63.55 | 64.37 | 72.48 |
| *FLEX-Mol-LLaMA judge scoring* | | | |
| Best-of-$N$ ($N$=16) | 68.85 | 69.83 | 77.49 |
| Preference Optimization | **76.41** | **75.92** | **80.10** |

Figure 4: (*Left*) Accuracy (%) trends on the parallel artificial membrane permeability assay (PAMPA; [64]) task with different judgment scores. (*Middle*) Accuracy trends on the number of sampled responses in best-of-$N$ sampling. (*Right*) Performance comparison with reward-guided Mol-LLaMA. We report accuracy with prompt styles of default, CoT, and task information, as described in [27].

| | Reason. | MLLM-as-a-Judge | | VL-Reward | MJ-Bench |
|---|---|---|---|---|---|
| | | Score (↑) | w. Tie (↑) | Overall (↑) | Safety (↑) |
| VL-7B | ✗ | 0.290 | 0.521 | 39.28 | 28.42 |
| | ✓ | **0.332** | **0.538** | **48.60** | **57.51** |
| Omni-7B | ✗ | 0.224 | 0.486 | 39.92 | 27.01 |
| | ✓ | **0.306** | **0.532** | **48.02** | **47.69** |

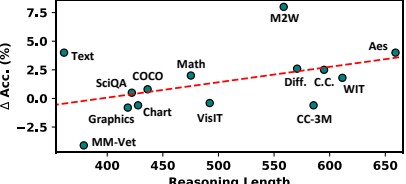

Figure 5: (*Left*) Performance comparison of FLEX-Judge with and without reasoning. (*Right*) Relationship between the average reasoning length of FLEX-VL-7B and the accuracy gain from reasoning over non-reasoning evaluation across subcategories in MLLM-as-a-Judge (Pair w. Tie).

adapters [26], molecular encoders, and Q-Formers [37]) unchanged. We refer to this model as **FLEX-Mol-LLaMA**.

We evaluate FLEX-Mol-LLaMA via two approaches. (1) Best-of-$N$ selector: FLEX-Mol-LLaMA guides best-of-$N$ sampling from Mol-LLaMA outputs, a method correlated with judge model performance [38]. (2) Direct preference optimization (DPO): Given a prompt $\mathbf{x}$, Mol-LLaMA generates two responses $\mathbf{y}_1$ and $\mathbf{y}_2$, and FLEX-Mol-LLaMA judge selects the preferred ($\mathbf{y}_w$) over the less preferred ($\mathbf{y}_l$) by score evaluation, forming triplets $(\mathbf{x}, \mathbf{y}_w, \mathbf{y}_l)$ [51]. These are then used to fine-tune Mol-LLaMA via DPO [56]. Strong downstream performance indicates that FLEX-Mol-LLaMA effectively curates high-quality preference data.

**Best-of-$N$ Selector.** We use FLEX-Mol-LLaMA to score the Mol-LLaMA's understanding of chemical property (permeability), following Kim et al. [27]. Our key findings include:

- **Score-Accuracy Correlation**: The judge scores strongly correlate with actual task performance (Figure 4; *left*), demonstrating that FLEX-Mol-LLaMA captures meaningful evaluation signals from molecular content and effectively detects high-quality responses.
- **Best-of-$N$ Sampling**: Selecting the highest-scoring ones among $N$ sampled responses yields a significant accuracy improvement (Figure 4; *middle*, up to 77.49% when $N = 16$), indicating that our judge model provides reliable preference signals.

**Reward-Guided Training via DPO.** In this sense, we further explore using FLEX-Mol-LLaMA judge as a reward source for DPO. We collect 4K samples of FLEX-Mol-LLaMA preferences $(\mathbf{x}, \mathbf{y}_w, \mathbf{y}_l)$ of the training set—covering chemical structures, properties, and biological features—and further fine-tune Mol-LLaMA. As shown in Figure 4 (*right*), DPO reaches up to 80.10%, surpassing the previous state-of-the-art. This result confirms that our judge model can also act as an effective reward model in underexplored modalities, enabling a scalable preference optimization. For the FLEX-Mol-LLaMA's reasoning and judgment results, refer to Appendix D.3.

## 5 Analysis

**Effect of Reasoning.** To investigate the impact of reasoning in FLEX-Judge, we compare it against a non-reasoning variant where the training segments `<think>` and `<answer>` are reversed—that is, the model is trained to generate the final answer first, followed by the reasoning. This answer-

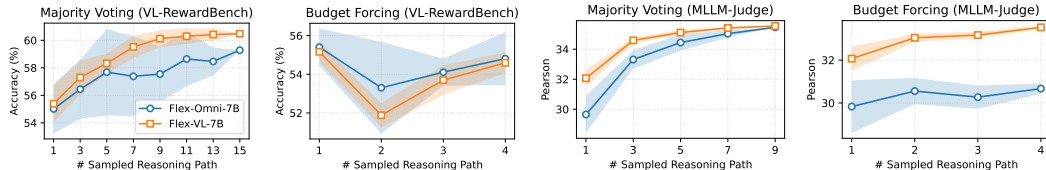

(a) Majority Voting (Pair)    (b) Budget Forcing (Pair)    (c) Majority Voting (Score)    (d) Budget Forcing (Score)

Figure 6: Inference-time scaling of FLEX-(VL/Omni)-7B. Unlike prior MLLM evaluators [38], our model supports parallel inference-time scaling via majority voting, showing consistent gains. Surprisingly, budget forcing also offers minor but consistent improvements in score-based evaluations.

then-reason format is commonly used in open-source datasets [76, 90]. As shown in Figure 5 (*left*), our proposed FLEX-Judge consistently outperforms this variant, demonstrating that reasoning-first evaluation leads to more robust and generalizable performance. Furthermore, in Figure 5 (*right*), we find that accuracy gains on MLLM-as-a-Judge correlate with the average reasoning length produced by FLEX-VL-7B. This suggests that more difficult tasks elicit deeper reasoning, highlighting the importance of reasoning capabilities in accurate evaluation. Notably, the M2W dataset [19], which involves complex website browsing scenarios and thus requires fine-grained reasoning paths, stands out with a larger performance gain than the general trend.

**Inference-time Scaling Works in FLEX-Judge.** We examine inference-time scaling techniques, including majority voting [69] and self-refinement through budget forcing [44, 50], to enhance the reasoning ability of our FLEX-Judge. In the VL-RewardBench (Reasoning) pairwise comparison, applying majority voting steadily improves performance, which is in stark contrast to prior work [38] that reported performance drops under inference-time scaling (Figure 6a). For budget forcing, we inject the keyword "Wait", shown to be effective in Muennighoff et al. [50]. Although performance drops after the first trial, it consistently improves in subsequent trials and nearly recovers to its original level (Figure 6b). In score-based evaluation, both methods show consistent gains (6c and 6d). These results highlight that, unlike existing MLLM-based judges, our evaluator benefits from increased inference-time computation, especially in reasoning-heavy tasks. Thanks to its reasoning-based training, our FLEX-Judge produces more diverse reasoning paths than prior non-reasoning evaluators, allowing inference-time scaling methods to be more effective.

**Data Quality vs. Modality Alignment.** To further assess the effectiveness of our cost-efficient approach, we compare FLEX-VL-7B with a variant trained on image-text evaluation pairs. Using the RLHF-V dataset [85], we curated 1K image-text pairs with reasoning-guided evaluation where FLEX-VL-7B judged the chosen response as better—without relying on explicit GPT-4o-annotated scores, *potentially* resulting in weaker quality than our original dataset (▲). As shown in Table 5, these

Table 5: Comparison of variants trained on potentially lower-quality image-text data. VL and HQ denote vision-language training and high-quality data.

| | VL | HQ | MLLM-as-a-Judge | | VL-Reward | GenAI |
|---|---|---|---|---|---|---|
| | | | Score (↑) | w. Tie (↑) | Overall (↑) | Video (↑) |
| VL-7B | ✓ | ▲ | 0.274 | 0.198 | 43.84 | 43.41 |
| | ✗ | ✓ | **0.332** | **0.538** | **48.60** | **44.78** |

variants consistently underperform FLEX-Judge, emphasizing that dataset quality outweighs modality-awareness.

## 6 Related Work

LLMs have increasingly been used as proxy evaluators in place of costly human annotators, a direction formalized under the LLM-as-a-Judge paradigm [8, 89], which has demonstrated strong alignment with human preferences [28, 60, 61, 63]. These model-based judgments can be leveraged in various downstream techniques such as best-of-$N$ selection [24] and preference-driven optimization using DPO [56, 65] or listwise reranking [43, 55]. However, most LLM-as-a-Judge approaches remain limited to text-only domains [36], while their multimodal variants (*e.g.,* ImageReward [78]) require large-scale, modality-specific annotations [6, 32, 76]. Recent work has highlighted the advantages of *reasoning*-guided supervision, training models with explanations or chain-of-thought rationales [7, 71, 72], to improve judgment quality and generalization [12]. Yet, collecting high-quality multimodal rationales is costly and especially difficult for underexplored modalities [11, 30,

88], limiting applicability to new domains. Our work addresses this gap by showing that textual reasoning alone can effectively train multimodal judge models to generalize across modalities, including tasks like molecular evaluation where no modality-specific preference data exists [27]. Additional related works are discussed in Appendix A.

## 7 Conclusion

In this work, we introduce **FLEX-Judge**, a reasoning-guided multimodal evaluator trained solely on textual preference explanations. By leveraging structured reasoning from a pretrained model, we have shown that FLEX-Judge generalizes to diverse modalities including image, video, audio, and molecules, without modality-specific supervision. Despite using far fewer annotations, it matches or outperforms state-of-the-art commercial APIs and open-source evaluators. These results highlight reasoning supervision as a scalable, cost-effective alternative for training general-purpose judge models in complex multimodal settings.

## Acknowledgements

SK, SC and SY were supported by Institute of Information & communications Technology Planning & Evaluation (IITP) grant funded by the Korea government (MSIT) (No. RS-2019-II190075, Artificial Intelligence Graduate School Program (KAIST); 10%) and the Institute of Information & communications Technology Planning & Evaluation (IITP) grant funded by the Korea government (MSIT) (No. 2022-0-00871, Development of AI Autonomy and Knowledge Enhancement for AI Agent Collaboration; 90%). We would like to thank Yeongjun Kim from the Georgia Institute of Technology for helpful discussions on the molecule evaluator.

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

# FLEX-Judge: Text-Only Reasoning Unleashes Zero-Shot Multimodal Evaluators

## Supplementary Material

## A  Additional Related Work

### A.1  (M)LLMs as Judges

Large language models (LLMs) have recently been explored as a cost-effective alternative to human raters, a paradigm referred to as LLM-as-a-Judge [89]. By encoding human-like reasoning and leveraging instruction prompts, these models can approximate human preference judgments in tasks such as summarization, dialogue, or code generation [29, 60, 61, 63]. Because human annotations are expensive and time-consuming to scale, the LLM-based evaluation promises a reusable, modular approach that can reduce reliance on direct human supervision. For instance, BoNBoN Alignment [24] explores using best-of-N selection guided by LLM-generated scores, while LRL [43] relies on LLM-based ranking signals to optimize model outputs at inference time.

However, most LLM-as-a-Judge approaches have focused on text-only use cases [36], and extending these models to multimodal content (*e.g.,* vision-language or speech tasks) remains challenging [10, 11, 68]. While multimodal variants do exist [76, 78], they often rely on large-scale, manually crafted, and modality-specific annotated datasets for training or fine-tuning [6, 77, 88], which becomes overly expensive at scale. For example, LLaVA-Critic [76] curates 46K images and 113K evaluation data by using GPT-4/4V on 8 multimodal datasets, as well as manually crafted prompt templates. Prometheus-Vision [32] involves a number of data processing steps, with 5K images and 15K customized score rubrics, and GPT-4V generating 150K response-feedback pairs.

Furthermore, the lack of high-quality, publicly available multimodal preference benchmarks makes it difficult to develop MLLM judge models [11, 30]. Our work addresses this gap by demonstrating that an MLLM can effectively judge multimodal outputs without requiring extensive modality-specific preference supervision. This approach suggests that LLMs as judge models can be extended to a wider range of modalities (*e.g.,* 3D point clouds [25], LiDAR [80], molecular [27, 83]) that have not yet been explored due to limited resources.

### A.2  Reasoning-Guided Reward Models

A growing body of research suggests that equipping reward models with the ability to reason—often captured through chain-of-thought prompts or explicit textual rationales—can significantly improve alignment with human preferences [7, 61, 72]. Rather than learning only from binary or scalar labels, these models benefit from learning how humans arrive at a decision, providing a more interpretable and generalizable internal mechanism for preference modeling [31, 57].

Recent advances in this area have shown that training on textual explanations can improve zero-shot and few-shot performance in downstream tasks requiring nuanced judgments [29, 60]. For example, Zheng et al. [89] find that when LLMs articulate the rationale behind their preference for one sample over another, they achieve higher consistency with human annotators. Building on the success of reasoning-based models in other domains, Zhang et al. [87] and Mahan et al. [45] introduced generative reward models, which significantly outperform non-reasoning judge models. More recently, JudgeLRM [12] demonstrates that reinforcement learning with reasoning-annotated data leads to substantial gains on reasoning-intensive evaluation tasks, significantly outperforming standard SFT-trained models. Concurrent with JudgeLRM, models such as J1 [73], DeepSeek-GRM [42], and RM-R1 [13] also adopt reinforcement learning frameworks to enhance the reasoning capabilities of reward models.

Despite these benefits, most reasoning-based reward modeling remains constrained to text-only applications, due in part to the higher cost and complexity of collecting multimodal rationales [11, 76]. In contrast, our work shows that *textual* reasoning supervision alone can be leveraged to enable robust multimodal evaluation. By training models on a small set of high-quality textual explanation data, we demonstrate an effective cross-modal generalization without the need for domain-specific

```
<|im_start|>system
You are a helpful assistant.  The assistant first performs a detailed,
step-by-step reasoning process in its mind and then provides the user with
the answer.  The reasoning process and answer are enclosed within <think>
</think> and <answer> </answer> tags, respectively, i.e., <think> detailed
reasoning process here, explaining each step of your evaluation for both
assistants </think><answer> answer here </answer>.  Now the user asks you
to judge the performance of two AI assistants in response to the question.
Score assistants 1-10 (higher=better).  Criteria includes helpfulness,
relevance, accuracy, and level of detail.  Avoid order, length, style or
other bias.  After thinking, when you finally reach a conclusion, clearly
provide your evaluation scores within <answer> </answer> tags, i.e., for
example, <answer>3</answer><answer>5</answer>
<|im_end|>
<|im_start|>user
 [Question]
{question}

 [Assistant 1's Answer]
{answer_1}

 [Assistant 2's Answer]
{answer_2}
<|im_end|>
<|im_start|>assistant
<think>
```

Figure 7: System prompt for JudgeLRM [12].

preference labels. This approach reduces annotation overhead and paves the way for more scalable, modality-agnostic judge models.

# B    Experimental Details

## B.1    Dataset Description

We first describe our **training dataset** (seed): To generate reasoning-based judgments, we use JudgeLRM-7B [12] to sample responses based on the given contexts from JudgeLM-100K [90], using the prompts shown in Figure 7 with a temperature of 0.1. Among the 100K samples, we filter out those with rating mismatches compared to the GPT-4o evaluation results provided in Zhu et al. [90], resulting in approximately 20K samples remaining after this process.

To construct the single-score format, we post-process 500 samples by truncating the reasoning and answer of `Assistant 2`, selecting cases where the reasoning length of `Assistant 1` exceeds 375 tokens.  For the pairwise format, we use the 500 longest reasoning samples where the combined length of both assistants' responses exceeds 750 tokens. For both pairwise and single-score formats, we randomly transform the grading scale from 1–10 to 1–5 by dividing the score by 2. We provide our final training dataset in the supplementary material.

We also describe all the benchmarks used in our experiments:

- **MLLM-as-a-Judge:** The MLLM-as-a-Judge benchmark [11] is introduced to specifically assess the judgment ability of MLLMs in the domain of image understanding across diverse scenarios. The benchmark consists of 14 datasets covering tasks such as image captioning, mathematical reasoning, text recognition, and infographic understanding.  In total, it includes 4,414 image-instruction pairs, curated to evaluate whether MLLMs can generalize their evaluative abilities across different modalities.
- **VL-RewardBench:** VL-RewardBench [38] is a diagnostic benchmark for evaluating vision-language models on multimodal understanding, hallucination detection, and complex reasoning.

Table 6: Description of the evaluation format, metrics used, and bias handling approach for evaluation benchmarks in Section 3.

| Benchmark | Eval. format | Metric | Bias handling |
|---|---|---|---|
| MLLM-as-a-judge | Single-score grading
Pairwise comparison
Batch-level ranking | Pearson correlation
Accuracy
Norm. Levenshtein distance | Average on 3 samples
Average on 3 samples, tie option [21]
Average on 3 samples |
| VL-RewardBench | Pairwise comparison | Accuracy | Average on 5 samples, random order [66] |
| MJ-Bench | Pairwise comparison | Accuracy | Order reverse [66], tie option [21] |
| GenAI-Bench | Pairwise comparison | Accuracy | - |
| Audio MOS/SS | Single-score grading | LCC & SRCC | System-level evaluation [68] |

It comprises 1,250 high-quality examples curated through an AI-assisted pipeline with human verification.

- **MJ-Bench:** MJ-Bench [14] is a benchmark designed to evaluate multimodal foundation models in the role of a judge for image generation tasks. It includes preference data across four key perspectives—text-image alignment, safety, image quality, and generation bias—each further divided into detailed subcategories. Each example consists of an instruction paired with a chosen and rejected image, enabling fine-grained assessment of model judgment.

- **GenAI-Bench:** GenAI-Bench [35] is a human-curated benchmark for evaluating image and video generation models across diverse composition skills. It includes 1,600 prompts from professional designers, avoiding subjective or inappropriate content, and covers over 5,000 human-verified skill tags. Unlike prior work, each prompt is annotated with multiple fine-grained tags. Specifically, GenAI-Bench supports image generation, image editing, and video generation tasks.

- **Audio MOS/SS Benchmark:** There is no unified, structured benchmark for audio evaluation tasks. Instead, Wang et al. [68] assessed speech quality and speaker similarity using four datasets: NISQA [49], BVCC [18], and SOMOS [46] for speech quality (712, 742, and 3,000 test samples, respectively), and VoxSim [2] for speaker similarity (2,776 test pairs). All datasets include human-annotated scores. For the MOS prediction task, auditory LLMs are asked to assign the MOS score on a scale from 1.0 to 5.0 for a given speech input. While Wang et al. [68] designed dataset-specific prompts to help models account for each dataset's unique standards, we evaluate models using a unified prompt format without dataset-specific tuning. For the SS prediction task, models rate the similarity between two speech samples on a scale from 1.0 to 6.0, where higher scores indicate greater speaker similarity. Since the human annotations of MOS and SS are averaged over multiple individuals, they are predicted in the form of floats.

## B.2 Training Details

Here, we describe the hyperparameters and implementation details for training FLEX-VL-7B and FLEX-Omni-7B. Using a *1K-sized* training dataset, we fine-tune Qwen2.5-VL-7B and Qwen2.5-Omni-7B with learning rates of $1 \times 10^{-5}$ and $7 \times 10^{-6}$, respectively. For both models, we use a batch size of 2 and a maximum sequence length of 4096 for a single epoch. Training is conducted on **2 NVIDIA A6000 GPUs**, taking approximately **1.5 hours** per run, which highlights cost-efficiency of our FLEX-Judge. For FLEX-Mol-LLaMA, we use the same hyperparameters as for FLEX-VL-7B.

## B.3 Details of Evaluation Protocol

All experiments in this work are designed to evaluate the quality of judge models, focusing on how closely their judgments align with human preferences when scoring or comparing AI-generated responses. Since our goal is to assess models that act as *evaluators*, we compare their decisions directly to human annotations across several evaluation formats.

Table 6 summarizes the evaluation formats, metrics, and bias-handling strategies used across all benchmarks used in our paper. Below, we elaborate on the evaluation settings:

- **Single-score Grading:** The judge assigns a scalar score to a single response. We evaluate performance using Pearson correlation (for vision-language tasks) or Spearman and LCC (for

```
<|im_start|>system
You are a helpful assistant.  The assistant first performs a detailed,
step-by-step reasoning process in its mind and then provides the user with
the answer.  The reasoning process and answer are enclosed within <think>
</think> and <answer> </answer> tags, respectively, i.e., <think> detailed
reasoning process here, explaining each step of your evaluation for an
assistant </think><answer> answer here </answer>.  Now the user asks you to
judge the performance of an AI assistant (multiple AI assistants) in response
to the question.  Score assistant 1-10 (higher=better).  Criteria includes
helpfulness, relevance, accuracy, and level of detail.  DO NOT assign the
same score to multiple assistants.  Avoid order, length, style or other
bias.  After thinking, when you finally reach a conclusion, clearly provide
your evaluation scores within <answer> </answer> tags, i.e., for example,
<answer>3</answer><answer>5</answer><answer>6</answer>
<|im_end|>
<|im_start|>user
 [Question]
{question}

 [Assistant 1's Answer]
{answer_1}

 [Assistant 2's Answer]
{answer_2}

 [Assistant 3's Answer]
{answer_3}

 [Assistant 4's Answer]
{answer_4}

<|im_end|>
<|im_start|>assistant
<think>
```

Figure 8: System prompt for single-score and batch-level ranking evaluations. The part colorized in red denotes the additional instruction used only for batch-level ranking evaluation.

audio), measuring the alignment between the judge's scores and human-annotated ratings. For audio benchmarks, evaluation is also conducted at the system level, averaging across all utterances from the same text-to-speech system.

- **Pairwise Comparison:** The judge selects the preferred response between two candidates. Accuracy is computed by counting the agreement between the judge and human preferences. We handle position bias by using randomized response orders [66] or incorporating the tie option [21], depending on the benchmark.

- **Batch-level Ranking:** The judge ranks multiple candidate responses (more than two) based on quality. Human-annotated rankings are treated as ground-truth, and we consolidate the ranking results into sequences (*e.g.,* ABCD ↔ CDAB) and measure their similarity using Normalized Levenshtein Distance [34].

**List of Prompts.** We provide our applied system prompts for diverse evaluations setup. For pairwise evaluation which is the most common setup in our experiments, we utilize the system prompt as in Figure 7 and post-process the results by comparing the scores provided from the FLEX-Judge. For single-score and batch-level ranking evaluations, our prompts are introduced in Figure 8, thanks to its applicability via post-processing. Despite minor variations among evaluation setups, models trained on datasets that do not incorporate the format diversity we introduce in Section 2.2 often fail to follow the given instructions. In contrast, our judge model—trained on data that reflects this format diversity—consistently adheres to the prompts across all setups.

```
<|im_start|>system
You are a helpful assistant.  The assistant first performs a detailed,
step-by-step reasoning process in its mind and then provides the user with
the answer.  The reasoning process and answer are enclosed within <think>
</think> and <answer> </answer> tags, respectively, i.e., <think> detailed
reasoning process here, explaining each step of your evaluation for an
assistant </think><answer> answer here </answer>.  Now the user asks you to
judge the performance of an AI assistants.  You have only FOUR Option:

Option 1.  Model A is better:  [[A>B]]
Option 2.  Model B is better:  [[B>A]]
Option 3.  Tie, relatively the same acceptable quality:  [[A=B=Good]]
Option 4.  Both are bad:  [[A=B=Bad]]

Assess the quality of generated videos.  Consider inappropriateness the
following sub-dimensions:  Alignment with editing prompt, Overedited,
Naturalness, Artifact, and Visual Appealing, are correctly represented.
Avoid order, length, style or other bias.  After thinking, when you finally
reach a conclusion, clearly provide your evaluation scores within <answer>
</answer> tags, i.e., for example, <answer>[[B>A]]</answer>.
<|im_end|>
<|im_start|>user
 [Question]
{question}

 [Assistant A's Video]
{video_1}

 [Assistant B's Video]
{video_2}

<|im_end|>
<|im_start|>assistant
<think>
```

Figure 9: System prompt for GenAI-Bench (edition) evaluation. We colorized the different parts in red compared to the training samples and observed that FLEX-Judge closely follows the given instructions, as shown in Figure 15.

For GenAI-Bench [35] and audio evaluations, which use different label formats (*e.g.,* [[A=B=Good]] or [[A=B=Bad]] for GenAI-Bench, and the first decimal point for audio evaluation) compared to our training samples, we use different prompts, as shown in Figure 9 and Figure 10. Thanks to our format-diversity-aware training dataset, FLEX-Judge can reliably follow instructions with high variation, as demonstrated in Figure 15 and https://flex-judge.github.io/. Further detailed prompts can be found in our provided code implementation.

## B.4  FLEX-Mol-LLaMA Judge

In this section, we present the training details of FLEX-Mol-LLaMA, a reasoning-augmented molecular judge model built on top of Mol-LLaMA [27]. Mol-LLaMA is a molecule-focused LLM, pretrained and fine-tuned on molecular understanding datasets. Its structure is based on the frozen LLaMA3.1-8B, with LoRA adapters, molecule encoders, and Q-Formers attached to enable effective encoding of 2D and 3D molecular structures.

To construct FLEX-Mol-LLaMA, we reuse the molecular encoders and adapter modules of Mol-LLaMA and fine-tune only the LLaMA3.1-8B backbone using the same text-only reasoning dataset employed for training FLEX-Omni-7B and FLEX-VL-7B. This results in a model that retains full molecular understanding while acquiring generalizable reasoning capabilities. LoRA modules are re-attached after fine-tuning, allowing us to preserve the domain-specific functionality of Mol-LLaMA while transforming it into a judge model.

```
<|im_start|>system
You are a helpful assistant.  The assistant first performs a detailed,
step-by-step reasoning process in its mind and then provides the user with
the answer.  The reasoning process and answer are enclosed within <think>
</think> and <answer> </answer> tags, respectively, i.e., <think> detailed
reasoning process here, explaining each step of your evaluation for an
assistant </think><answer> answer here </answer>.  Now the user asks you to
judge the performance of an audio generative AI assistant in response to the
question.  Listen to the generated speech audio, and score this speech on a
scale from 1.0 to 5.0 in FIRST DECIMAL. Consider the following criteria when
scoring:

1 - Very Bad:  The speech is very unnatural, has poor audio quality, and is
nearly impossible to understand.
2 - Poor:  The speech sounds unnatural and/or noisy.  Only a few words are
understandable.
3 - Fair:  The speech is somewhat unnatural or contains noticeable noise, but
the overall meaning is understandable.
4 - Good:  The speech is generally natural and clear, with most of the
content easy to understand.
5 - Excellent:  The speech is very natural, high in audio quality, and fully
intelligible.

Do NOT consider the content of the speech.  After thinking, when you finally
reach a conclusion, clearly provide your evaluation scores within <answer>
</answer> tags, i.e., for example, <answer>3.8</answer>.
<|im_end|>
<|im_start|>user
 [Question]
Generate clear, natural, and understandable high-quality speech audio.

 [Assistant's Answer]
Here is the speech I generated:  {audio}

<|im_end|>
<|im_start|>assistant
<think>
```

Figure 10: System prompt for speech quality assessment. We colorized the different parts in red compared to the training samples.

### B.4.1 Best-of-N Sampling

We first evaluate FLEX-Mol-LLaMA as a reward model for inference-time scaling. Specifically, we apply it to the best-of-$N$ sampling setup, where $N$ number of responses are sampled from the base Mol-LLaMA, and the best one is selected using FLEX-Mol-LLaMA's predicted scores. Importantly, each response includes not only the Mol-LLaMA's final prediction label (*i.e.,* "high permeability" or "low-to-moderate permeability"), but also its accompanying analysis and explanation for the prediction. Scores assigned by FLEX-Mol-LLaMA generally range between 6.0–9.5 and show a strong correlation with downstream task accuracy (see Figure 4; *left*), suggesting that the model effectively distinguishes between higher- and lower-quality outputs.

Since there exist responses that receive identically best scores, we repeat the sampling and selection process over 10 random trials and report the average performance. As shown in Figure 4 (*middle*), increasing $N$ consistently improves accuracy, validating that FLEX-Mol-LLaMA provides reliable, fine-grained reward signals.

### B.4.2 DPO Training

Beyond inference-time selection, we also use FLEX-Mol-LLaMA as a reward model for DPO to further fine-tune Mol-LLaMA. For this, we curate a DPO training dataset from Mol-LLaMA's instruc-

tion tuning corpus, which consists of two main types: (1) detailed chemical structural descriptions, and (2) structure-to-feature relationship explanations covering both chemical and biological attributes. We excluded the multi-turn conversation type.

For each query $\mathbf{x}$, we sample two responses from Mol-LLaMA using different decoding temperatures, 0.8 and 1.2. We use FLEX-Mol-LLaMA to compare the two and include the example in the DPO training set only if the response from temperature 0.8 receives a higher score than the one from 1.2. Also, since there is a position bias [66] in pairwise comparisons, we flip the order of the two responses in prompt and evaluate again, retaining only those pairs where the winning response remains consistent. Consequently, we construct 4,253 high-quality preference triplets $(\mathbf{x}, \mathbf{y}_w, \mathbf{y}_l)$.

Fine-tuning Mol-LLaMA with this DPO dataset results in a substantial accuracy boost on the downstream permeability prediction task. As shown in Figure 4 (*right*), the final model achieves up to 80.10% accuracy, surpassing prior models by a large margin. This result confirms that FLEX-Mol-LLaMA provides not only reliable evaluation at inference but also effective supervision for preference-based training in specialized, underexplored domains.

# C   Additional Results

## C.1   Additional Benchmarks

We extend our evaluation to include results on two recent and comprehensive MLLM evaluation benchmarks: Multimodal RewardBench [82], which focuses on image understanding, and JudgeAnything [54], which covers wide range of any-to-any tasks. JudgeAnything includes not only image/video/audio understanding (I → T, V → T, A → T) but also image/video/audio generation from text (T → I, T → V, T → A) and more complex tasks like video-to-audio (V → A) or audio-visual-to-text (V+A → T).

The results are summarized in Table 7 and Table 8, respectively. We note that FLEX-VL-7B is not capable of evaluating audio-related tasks. Across both benchmarks, FLEX-Judge achieves comparable performance to proprietary models while consistently outperforming existing open-source baselines.

Table 7: Comparison of MLLM evaluator performance on Multimodal RewardBench. $\diamond$: results from the original work [82].

| Model | General | | Knowledge | Reasoning | | Safety | VQA | Overall |
|---|---|---|---|---|---|---|---|---|
| | Correctness | Preference | | Math | Coding | | | |
| GPT-4o$^\diamond$ | **0.626** | **0.690** | **0.720** | **0.676** | **0.621** | **0.748** | **0.872** | **0.715** |
| LLaMA-3.2-90B-Vision$^\diamond$ | 0.600 | 0.684 | 0.612 | 0.563 | 0.531 | 0.520 | 0.771 | 0.624 |
| LLaVA-1.5-13B$^\diamond$ | 0.533 | 0.552 | 0.505 | 0.535 | 0.493 | 0.201 | 0.518 | 0.489 |
| FLEX-Omni-7B | 0.616 | 0.612 | 0.657 | 0.625 | 0.582 | 0.362 | 0.777 | 0.631 |
| FLEX-VL-7B | 0.620 | 0.618 | 0.630 | 0.677 | 0.550 | 0.732 | 0.837 | 0.686 |

Table 8: Comparison of MLLM evaluator performance on JudgeAnything within the *Overall* setting and pair comparison with tie. $\diamond$: results from the original work [54].

| Model | Multimodal Understanding | | | | | Multimodal Generation | | | | | | | | | | Overall |
|---|---|---|---|---|---|---|---|---|---|---|---|---|---|---|---|---|
| | T→T | I→T | V→T | A→T | V+A→T | T→I | T→V | T→A | I→I | I→V | I→A | V→V | V→A | A→V | A→A | |
| GPT-4o$^\diamond$ | **52.5** | 73.0 | 58.0 | **69.5** | **53.0** | **52.5** | 56.0 | 26.0 | 49.0 | 42.0 | 68.0 | 78.0 | 38.0 | 58.5 | 57.5 | 55.4 |
| Gemini-1.5-Pro$^\diamond$ | 49.0 | **79.0** | 57.5 | 68.5 | 48.5 | 52.0 | 56.0 | 38.0 | **57.0** | 39.0 | 69.0 | **88.5** | 33.0 | 47.0 | **67.5** | **56.6** |
| Gemini-2.0-Flash$^\diamond$ | 43.5 | 67.5 | **59.0** | 69.0 | 51.5 | 46.5 | **58.0** | 50.5 | 51.0 | 43.5 | 57.5 | 77.0 | 41.0 | 58.0 | 38.5 | 54.1 |
| Qwen2.5-Omni-7B | 35.5 | 50.5 | 42.9 | 51.5 | 44.5 | 36.0 | 31.5 | 39.5 | 29.5 | 40.0 | 32.5 | 34.5 | 35.5 | 53.0 | 40.5 | 39.8 |
| Qwen2.5-VL-7B | 36.5 | 33.5 | 33.5 | - | - | 36.5 | 34.5 | - | 27.0 | 34.0 | - | 34.0 | - | - | - | - |
| FLEX-Omni-7B | 46.5 | 70.0 | 53.5 | 67.0 | 52.0 | 49.0 | 56.0 | 40.5 | 47.0 | **45.0** | 72.5 | 78.0 | 44.0 | **70.0** | 38.0 | 55.3 |
| FLEX-VL-7B | 45.0 | 68.5 | 53.5 | - | - | 43.5 | 54.5 | - | 47.5 | 42.0 | - | 70.5 | - | - | - | - |

## C.2   Scaling Law for FLEX-Judge

To examine how model scale affects judge performance, we conduct an additional experiment using a smaller LLM backbone, where results are found in Table 9. Specifically, we train FLEX-VL-3B

by fine-tuning Qwen2.5-VL-3B on our curated seed dataset. Despite its significantly smaller size, FLEX-VL-3B also demonstrates reasoning capabilities and generalizes across modalities. However, it consistently underperforms compared to our base 7B model, indicating that larger-scale LLM-based models are better equipped to internalize reasoning patterns and serve as more reliable judges. These results suggest that while reasoning-guided supervision is effective even at smaller scales, model capacity remains an important factor in achieving high-quality, generalizable judgments. Meanwhile, despite its slightly lower performance compared to the 7B model, FLEX-VL-3B shows comparable performance to LLaVA-1.6-34B on MLLM-as-a-Judge and both Qwen2.5-VL-7B and LLaVA-NeXT on GenAI-Bench.

Table 9: Comparison of FLEX-Judge performance across different MLLM sizes. ♠: results from Table 1. ♡: results from Table 2. ◇: results from Table 3. †: 32B model was trained with LoRA [26].

| Model | Size | MLLM-as-a-Judge | | | | VL-Reward | | | GenAI-Bench | | |
|---|---|---|---|---|---|---|---|---|---|---|---|
| | | Score (↑) | w. Tie (↑) | w.o. Tie (↑) | Batch (↓) | General (↑) | Hallu. (↑) | Reason. (↑) | Image (↑) | Edition (↑) | Video (↑) |
| LLaVA-1.6♠ | 34B | 0.184 | 0.460 | 0.648 | 0.501 | - | - | - | - | - | - |
| Qwen2-VL♡ | 72B | - | - | - | - | 38.1 | 32.0 | 61.0 | - | - | - |
| LLaVA-NeXT◇ | Unk. | - | - | - | - | - | - | - | 22.65 | 25.35 | 21.70 |
| Qwen2.5-VL◇ | 7B | - | - | - | - | - | - | - | 31.93 | 38.63 | 37.61 |
| FLEX-VL | 3B | 0.176 | 0.493 | 0.650 | 0.478 | 46.39 | 36.85 | 58.08 | 36.71 | 33.62 | 42.93 |
| | 7B | 0.332 | 0.538 | 0.655 | **0.426** | 46.11 | **43.39** | 62.87 | 43.32 | 47.41 | 44.78 |
| | 32B† | **0.361** | **0.587** | **0.716** | 0.432 | **54.52** | 41.92 | **64.46** | **44.32** | **53.54** | **47.33** |

## C.3 Reliability of FLEX-Judge

**Length Bias.** In Chen et al. [11], models such as GPT-4V [1] and Gemini [62] tend to favor longer answers over concise yet correct ones, exhibiting a phenomenon known as verbosity bias [89]. In contrast, our FLEX-Omni-7B and FLEX-VL-7B demonstrate length preferences that are more consistent with human evaluators, unlike previous judge models reported in Chen et al. [11]. As illustrated in Figure 11, our models do not systematically prefer lengthy answers, but instead align well with human judgments regardless of response lengths. This suggests that FLEX-Judge can serve as a reliable judge.

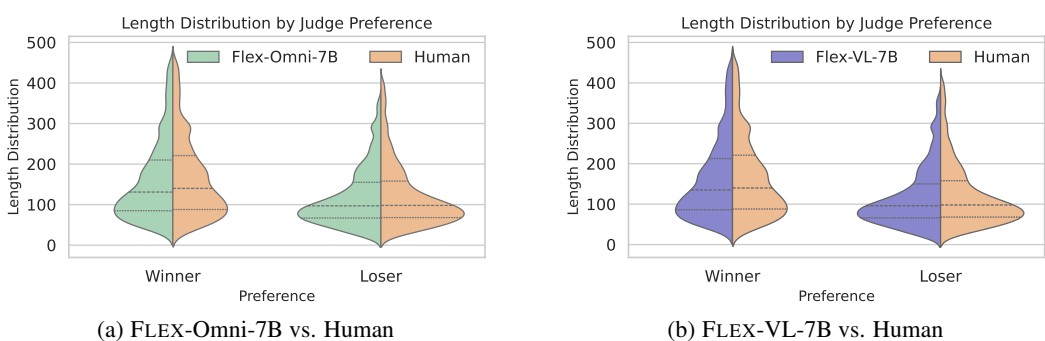

(a) FLEX-Omni-7B vs. Human          (b) FLEX-VL-7B vs. Human

Figure 11: Examination on length bias of FLEX-Omni-7B and FLEX-VL-7B compared to human evaluators in pairwise comparisons (excluding ties) on the MLLM-as-a-Judge benchmark [11].

**Position bias.** Models as judges consistently favor answers in specific positions, often influenced by training data that typically place correct responses at the beginning or end of prompts [89]. We observed similar behavior in our FLEX-Judge during pairwise comparison evaluations. In Figure 12, we examine the behavior of our judge models against human preferences. While human evaluators show less bias towards the first or second response and frequently opt for the Tie option when appropriate, our judge models, particularly FLEX-Omni-7B, tend to favor the first response more often. Additionally, our models are less likely to output Tie judgments, instead preferring one over another. As detailed in Table 6, we address the positional bias via randomly changing the response orders.

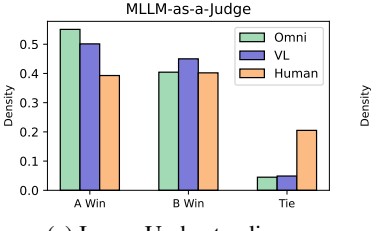
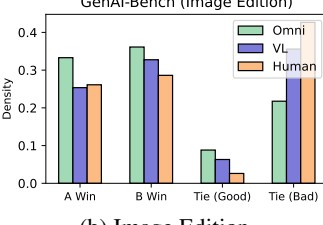
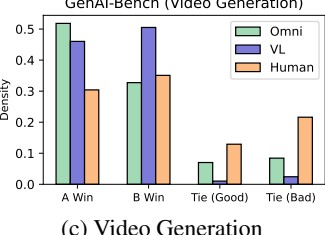

|              (a) Image Understanding              |              (b) Image Edition              |              (c) Video Generation              |

Figure 12: Examination on position bias of FLEX-Omni-7B and FLEX-VL-7B compared to human evaluators in pairwise comparisons (including ties). "A Win" indicates that the model preferred the first response, and "B Win" indicates preference for the second response.

## C.4 Text Judgment Performance

We also evaluate FLEX-Judge on text-only assessment tasks, with results shown in Table 10. Interestingly, both FLEX-Omni-7B and FLEX-VL-7B outperform the base judge model, JudgeLRM-7B, in terms of judgment accuracy—despite being trained on the JudgeLRM's response data. This suggests that our high-quality dataset curation (as discussed in Section 2.2) may lead to stronger textual reasoning performance.

While increasing the size of the training set could further improve test-time judgment accuracy, our primary focus is on multimodal generalization. Moreover, as discussed in Section 2.2, we observe catastrophic forgetting on the non-textual modalities when training with excessive text data. To balance effectiveness and modality retention, we limit the training set to a 1K-sized text corpus throughout our experiments.

Table 10: Comparison of (M)LLM evaluator performance on JudgeLM [90] and PandaLM [70]. ♠: results from Chen et al. [12]. †: re-implemented results.

| Model | JudgeLM (GPT-4o as Ground-Truth) | | | | PandaLM (Human as Ground-Truth) | | | |
|-------|-----------|-----------|--------|-------|-----------|-----------|--------|-------|
|       | Agreement | Precision | Recall | F1    | Agreement | Precision | Recall | F1    |
| GPT-3.5♠ | 73.83 | 70.70 | 52.80 | 52.85 | 62.96 | 61.95 | 63.59 | 58.20 |
| GPT-4♠ | - | - | - | - | 66.47 | 66.20 | 68.15 | 61.80 |
| PandaLM-7B♠ | 68.61 | 40.75 | 38.82 | 39.41 | 59.26 | 57.28 | 59.23 | 54.56 |
| Auto-J-13B♠ | 74.86 | 61.65 | 57.53 | 58.14 | - | - | - | - |
| JudgeLM-33B♠ | **89.03** | 80.97 | 84.76 | 82.64 | 75.18 | 69.30 | 74.93 | 69.73 |
| Qwen2.5-7B-Instruct♠ | 76.85 | 78.71 | 77.85 | 78.28 | 63.96 | 61.95 | 67.61 | 59.81 |
| JudgeLRM-7B♠ | 83.74 | **85.84** | 83.65 | 84.73 | **78.28** | **74.90** | **75.74** | **75.05** |
| JudgeLRM-7B† | 82.26 | 84.86 | 82.47 | 83.64 | 76.41 | 71.41 | 71.30 | 71.16 |
| FLEX-Omni-7B | 84.12 | 80.95 | **93.51** | **86.78** | 76.24 | 73.38 | 71.15 | 71.74 |
| FLEX-VL-7B | 84.21 | 82.65 | 90.06 | 86.65 | 76.84 | 72.15 | 70.06 | 70.74 |

## C.5 Non-judgment Visual-Language Task Performance

To further examine potential catastrophic forgetting or degradation of multimodal abilities, we also evaluate our model on standard visual question answering (VQA) tasks that are unrelated to judgment, specifically TextVQA [59] and OK-VQA [47]. As shown in Table 11, FLEX-VL-7B exhibits only a slight decrease on TextVQA and even outperforms the base model on OK-VQA. These results suggest that our well-structured methodology preserves the multimodal capabilities of the base model and does not cause forgetting.

Table 11: Evaluation on standard visual question answering tasks.

| Model | TextVQA | OK-VQA |
|-------|---------|--------|
| Qwen2.5-VL-7B | 80.80 | 67.46 |
| FLEX-VL-7B | 80.31 | 72.10 |

# D Qualitative Examples

We provide qualitative examples of FLEX-Judge's reasoning and judgments across different modalities. These examples illustrate how the model understands, compares, and scores AI-generated outputs as well as multimodal inputs using structured textual reasoning.

## D.1 Image Understanding and Generation Tasks

We showcase examples of the image understanding and generation tasks with vision-language benchmarks. All judgments are made by FLEX-VL-7B.

Figure 13 illustrates an image understanding task, involving OCR of a Hindi phrase. Both assistants output similar-looking responses, but FLEX-Judge correctly recognizes the image content by referring to an "unexpected election" rather than a "financial crisis". It assigns a significantly higher score to Assistant 2, whose response accurately reflects the OCR content.

Figure 14 presents a quality assessment task of generated images. Both assistants generate images, and FLEX-judge evaluates them based on clarity, focus, and presence of artifacts. Assistant 1 receives

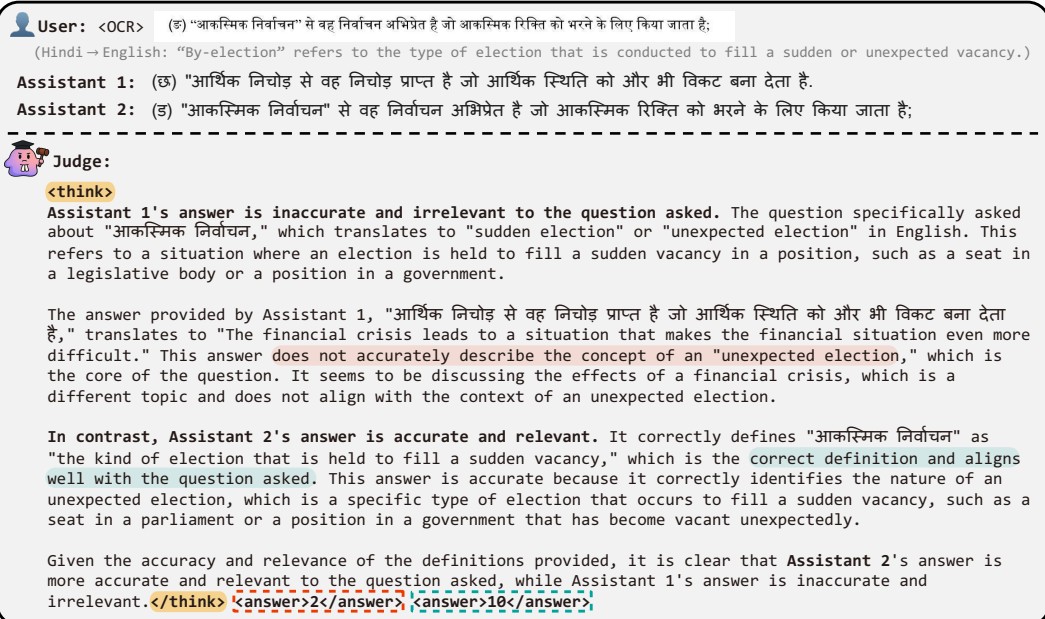

Figure 13: Reasoning process of FLEX-Judge on the OCR task (VL-RewardBench).

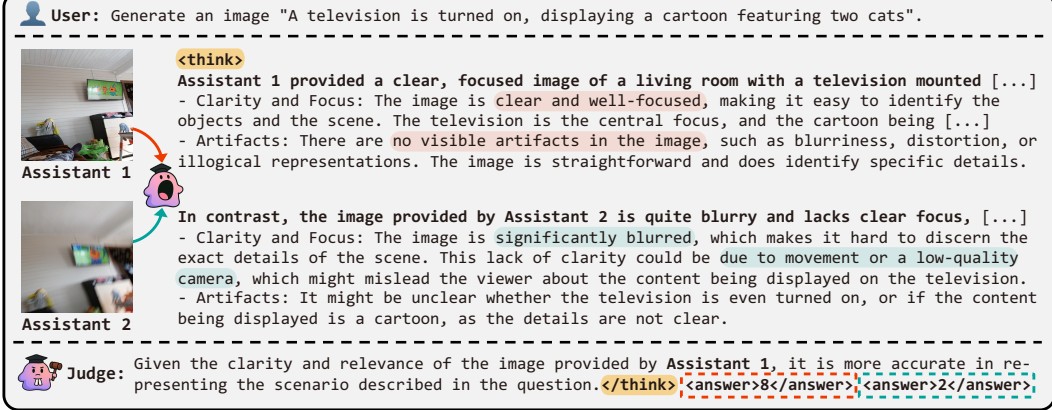

Figure 14: Reasoning process of FLEX-Judge on the image quality assessment task (MJ-Bench).

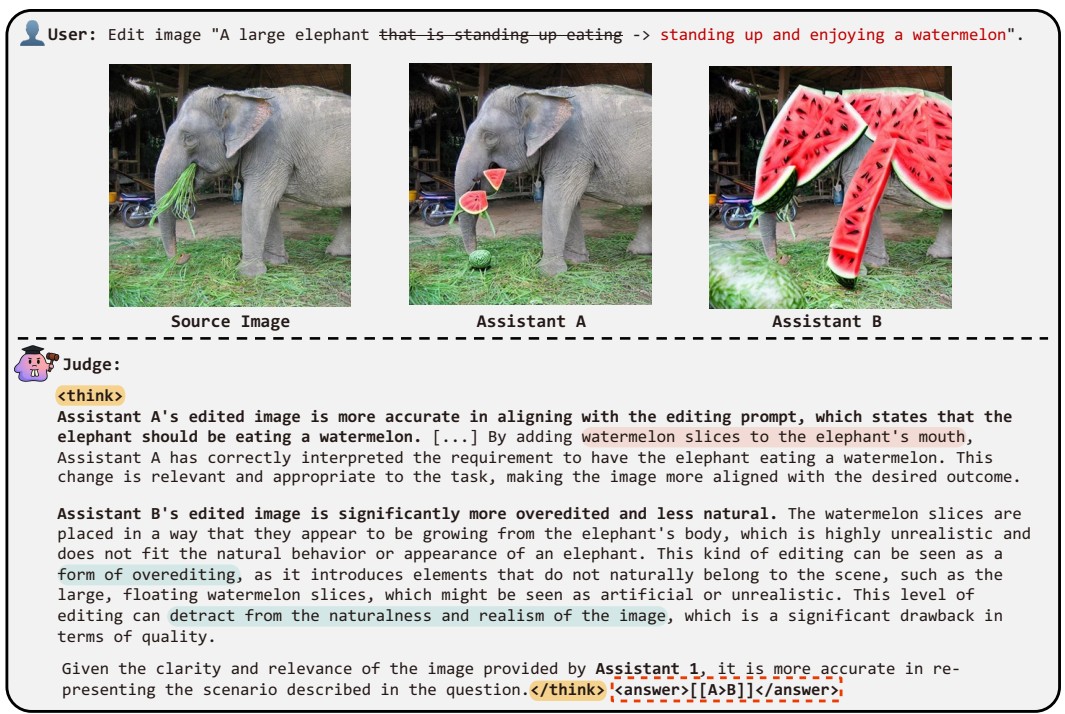

Figure 15: Reasoning process of FLEX-Judge on the image editing task (GenAI-Bench).

a higher score due to its "clear and well-focused" image with "no visible artifacts", whereas Assistant 2's image shows blurs, which could be "due to movement or a low-quality camera".

Figure 15 shows an image editing task, where the judge assesses whether the edited image by each assistant aligns with the revised prompt. FLEX-Judge identifies that Assistant B's output exhibits "overediting", detracting from the "naturalness and realism", and thus assigns it a lower score.

These examples highlight the FLEX-Judge's ability to explain its preferences based on semantic accuracy, relevance, and consistency with visual content, even in fine-grained evaluation scenarios.

## D.2 Video and Audio Tasks

Since we cannot include video and audio content directly into the paper, we provide full qualitative examples—including input prompts, AI responses, and FLEX-Judge's reasoning—for video and audio evaluation tasks at the following anonymized project page.[2]

## D.3 Molecular Tasks

Figure 16 and Figure 17 illustrate the reasoning and judgment outputs of FLEX-Mol-LLaMA, our molecular judge model. Figure 16 presents two examples from the PAMPA prediction task, the primary evaluation task for Mol-LLaMA. These samples were used in the best-of-N sampling experiment (Section 4). We display both a low-scoring and a high-scoring response examples, along with FLEX-Mol-LLaMA's reasoning, to highlight its ability to distinguish response quality based on molecular analysis. Figure 17 shows a pairwise judgment example from Mol-LLaMA's instruction tuning set, which includes prompts about biological property prediction. Here, we sample two responses from Mol-LLaMA and use FLEX-Mol-LLaMA to evaluate and score both. These pairwise preferences are then used to construct training triplets for DPO, guiding further alignment of Mol-LLaMA with high-quality response patterns.

[2]https://flex-judge.github.io/

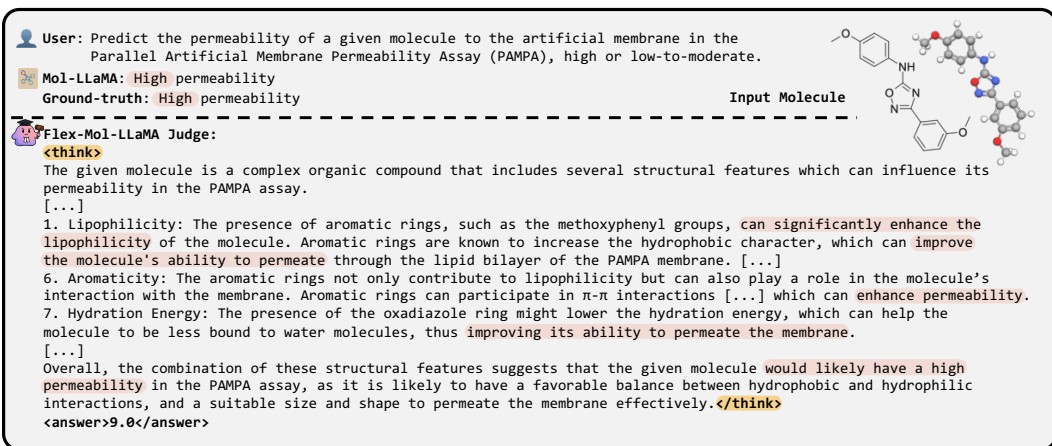

Figure 16: Reasoning process of FLEX-Mol-LLaMA judge on the PAMPA task.

## E   Limitations

We have demonstrated that textual reasoning alone can effectively train multimodal judge models that generalize across modalities. While training only the LLM part is an innovative idea, it consists of a pitfall: the underlying LLM must possess (or at least can learn) sufficient reasoning capability. Our method is efficient and broadly applicable, but it implicitly requires that the backbone model be capable of generating coherent, structured reasoning.

We observed this limitation when attempting to extend our approach to 3D-LLM [25], which can process and encode 3D point clouds using Flan-T5-XL [17] as its language backbone. Despite the model's 3D understanding, its limited context window (512 tokens) and lack of strong reasoning pretraining made it unsuitable for generating high-quality reasoning data. As a result, we were unable to successfully train a FLEX-3D-LLM judge using our framework. This highlights a key constraint of our approach: it is less effective when applied to MLLMs built on weak or constrained LLMs that lack the capacity for textual reasoning.

While FLEX-Judge naturally inherits the capabilities of its MLLM backbone, this is an inherent assumption shared by all LLM-as-a-Judge paradigms. Our key contribution lies not in developing a better backbone but in demonstrating that minimal textual supervision can yield strong cross-modal evaluation with a fixed MLLM, eliminating the need for expensive modality-specific training.

**Input Molecule (IUPAC name):** (2S)-5-amino-2-[[(2S)-2-amino-3-phenylpropanoyl]-methylamino]-N-naphthalen-2-ylpentanamide

👤 **User:** Elaborate in detail on the biological attributes of the provided compound, explaining how its primary substructures are linked to its properties.

🧬 **Mol-LLaMA (response 1):**
The given molecule, N-Acetylfentanyl, is a synthetic opioid analogue closely related to fentanyl. It is characterized primarily by the presence of several key substructures that contribute to its biological properties, particularly its potent opioid-like effects.

The core substructure of the given molecule includes an amide linkage, which is a feature common in many biologically active peptides and proteins. This linkage in the molecule is involved in forming a stable bond that allows the molecule to resist metabolic degradation, thereby enhancing its duration of action in the body. [...]

🧬 **Mol-LLaMA (response 2):**
The given molecule is a derivative of L-lysine connected to two napsigogen molecules. Let's delve into the biological properties by analyzing the core substructures:

1. L-lysine Backbone: L-lysine is an amino acid with a primary amine group, a carboxylic acid group, and a side chain bearing another amino group. Its configuration lends it essential roles in protein synthesis, serving as an acid, a base, and a player in enzymatic binding sites. With its charged groups, L-lysine can interact and stabilize protein structures through ionic bonds and hydrogen bonds. In the given molecule, this structure likely provides scaffold stability and contributes to its biochemical interactions.

2. Naphthyl Groups: Being a naphthyl amino acid implies the incorporation of naphthalene, an aromatic hydrocarbon. Two naphthyl groups are mentioned, each acting as a large, planar, aromatic moiety that can engage in pi-stacking interactions. [...]

- - - - - - - - - - - - - - - - - - - - - - - - - - - - - - - - - - - - - - - - - - - - - - - - - -

🤖 **Flex-Mol-LLaMA Judge:**
**<think>**
**Assistant 1 provided a very accurate and relevant explanation of the biological properties of N-Acetylfentanyl,** which is a synthetic opioid analogue. The core substructures in this molecule, such as the amide linkage, the chiral amino group, and the aromatic phenyl and naphthyl groups, are crucial in defining its biological activity.

1. Amide Linkage: This is a key feature that contributes to the molecule's stability and resistance to metabolic degradation. Amide bonds are often found in biologically active compounds because they are relatively stable and can help the molecule maintain its structure in the presence of enzymes that might otherwise break it down. This stability is important for the prolonged action of the molecule in the body, which is a typical characteristic of opioid analogues. [...]

**In contrast, the description provided by Assistant 2 is less accurate and relevant to the biological properties of N-Acetylfentanyl.** The core substructures described in the answer, such as the L-lysine backbone and the naphthyl groups, do not align well with the typical characteristics of opioid analogues. L-lysine is an amino acid, which is not a typical feature in opioid structures, and the presence of naphthyl groups might be more relevant to a different type of molecule or drug, such as those involved in protein interactions or specific enzyme inhibitors. This might lead to a different set of biological interactions and effects, which are not typical for opioid analogues like N-Acetylfentanyl.

Overall, the explanation provided by Assistant 1 is more aligned with the known properties and mechanisms of action of N-Acetylfentanyl, which is consistent with its role as a potent opioid analogue.**</think>**
**<answer>9</answer>** **<answer>4</answer>**

Figure 17: Reasoning process of FLEX-Mol-LLaMA, judging the preference of two Mol-LLaMA responses on biological attributes prediction data. Here, the response 1 is preferred (score 9) against the response 2 (score 4).

