# OpenReview forum: "Flex-Judge: Text-Only Reasoning Unleashes Zero-Shot Multimodal Evaluators"
_NeurIPS.cc/2025/Conference — NeurIPS 2025 poster_

### Official Review · Reviewer_VVyF · 2025-06-09

**Clarity:** 3
**Significance:** 2
**Originality:** 2
**Rating:** 4
**Confidence:** 3

**Summary:**

This paper introduces KIRBY-Judge, a reasoning-guided multimodal evaluator that uses minimal textual reasoning data to asses model outputs across various modalities like text, images, and videos. The key insight is that structured textual explanations capture generalizable decision-making patterns, which enables effective transfer to multimodal evaluation without extensive modality-specific training. Empirical results show that KIRBY-Judge achieves performance competitive with or better than existing methods, even in low-resource domains.

**Questions:**

- Could the authors elaborate more on the intuition of generalizing from textual reasoning data to multimodal judge capability?
- How exactly was the seed training data curated?  Could the authors provide some examples of their data?  Also, the data is mainly synthetic - could the authors elaborate on how to ensure the synthetic data is high quality and not collapsed?

**Ethical Concerns:**

["NO or VERY MINOR ethics concerns only"]

**Final Justification:**

Issues resolved

**Limitations:**

Yes

**Quality:**

3

**Strengths And Weaknesses:**

Strengths
- The method is solid.  The approach is modality-agnostic, and once trained, the KIRBY-Judge can be applied to diverse evaluation tasks (vision-language tasks, audio quality scoring, molecular structure, etc.)
- Overall, good experiments and execution.  Evaluation is conducted across diverse tasks, such as Image Understanding, Image Editing, Video Generation, Audio Understanding, and Molecule Evaluator.

Weakness
- It would be better if experimental results (Table 1, 2, 3) include stronger baselines like latest GPT4-o, o3 and Gemini 2.5 Pro.
- Misses some related work. For example, there are various multimodal judge or reward benchmark works worth discussing or evaluating on: video LM as judge (https://arxiv.org/abs/2503.05977) and Multimodal RewardBench (https://arxiv.org/abs/2502.14191), in addition to MLLM-as-a-Judge and VL-RewardBench, etc. discussed in the paper.

---

> ### Author Rebuttal · Authors · 2025-07-30
>
> We thank the reviewer for their constructive feedback. Below we address each concern in detail. We look forward to any further discussions.
>
> > ***Q1. [Inclusion of latest baseline models]*** It would be better if the experimental results included stronger baselines such as the latest Gemini 2.5 Pro.
>
> ***A1.*** Thank you for the suggestion. We have presented Gemini-2.5-Pro results in the table below. The latest Gemini-2.5-Pro model shows mixed superiority over previous versions.
>
> | Model             | MLLM-as-a-judge (Score, Ave.) | MLLM-as-a-judge (Pair w Tie, Ave.) | MLLM-as-a-judge (Pair w.o Tie, Ave.) | MLLM-as-a-judge (Batch, Ave.) $\downarrow$ | VL-RewardBench (General) | VL-RewardBench (Hallu.) | VL-RewardBench (Reason.) | GenAI-Bench (Image) | GenAI-Bench (Edition) | GenAI-Bench (Video) |
> |-|-|-|-|-|-|-|-|-|-|-|
> | Gemini-2.5-Pro    | 0.390 | 0.556 | 0.668 | 0.512 | 44.28 | 49.13 | 53.01 | 47.55 | 65.51 | 50.33 |
> | Gemini-1.5/1.0-Pro | 0.304 | 0.509 | 0.615 | 0.432 | 50.80 | 72.50 | 64.20 | 44.67 | 55.93 | 46.21 |
> | Kirby-Omni-7B     | 0.306 | 0.532 | 0.650 | 0.425 | 47.01 | 42.72 | 61.08 | 38.15 | 46.73 | 37.10 |
> | Kirby-VL-7B       | 0.332 | 0.538 | 0.655 | 0.426 | 46.11 | 43.39 | 62.87 | 43.32 | 47.41 | 44.78 |
>
> However, we would like to emphasize that our ultimate goal is not to surpass proprietary models on conventional benchmarks, but to enable capabilities that they currently lack. Kirby-Judge is designed to be adaptable to new, underexplored domains, such as molecular tasks, where existing models cannot be directly applied—which we detail further in ***Section 4 (Broader Impact)***.
>
> &nbsp;
>
> > ***Q2. [Additional related work]*** Misses some related work. For example, there are various multimodal judge or reward benchmark works worth discussing or evaluating on: Multimodal RewardBench, in addition to MLLM-as-a-Judge and VL-RewardBench, etc. discussed in the paper.
>
> ***A2.*** Thank you for the suggestion. We will include the mentioned works and discussion in our related work section. Additionally, we have extended our evaluation by including results on a recent MLLM evaluation benchmark, Multimodal RewardBench (Yasunaga et al., 2025) and JudgeAnything (Pu et al., 2025).
>
> Multimodal RewardBench (Yasunaga et al., 2025) focuses on image understanding, while JudgeAnything (Pu et al., 2025) covers a wide range of any-to-any tasks. JudgeAnything includes not only image/video/audio understanding (I → T, V → T, A → T) but also image/video/audio generation (T → I, T → V, T → A), image editing (I → I), image-to-video (I → V), and image-to-audio (I → A) tasks.
>
> **Multimodal RewardBench**:
>
> | Model | Overall | Correctness | Preference | Knowledge | Math  | Coding | Safety | VQA   |
> |-|-|-|-|-|-|-|-|-|
> | GPT-4o | 0.715 | 0.626 | 0.690 | 0.720 | 0.676 | 0.621 | 0.748 | 0.872 |
> | Llama-3.2-90B-Vision-Instruct | 0.624 | 0.600 | 0.684 | 0.612 | 0.563 | 0.531 | 0.520 | 0.771 |
> | Llava-1.5-13B | 0.489 | 0.533 | 0.552 | 0.505 | 0.535 | 0.493 | 0.201 | 0.518 |
> ||||||||||
> | Kirby-VL-7B | 0.686 | 0.620 | 0.618 | 0.630 | 0.677 | 0.550 | 0.732 | 0.837 |
> | Kirby-Omni-7B | 0.631 | 0.616 | 0.612 | 0.657 | 0.625 | 0.582 | 0.362 | 0.777 |
>
> **JudgeAnything**:
>
> | Model | I → T | V → T | A → T | T → I | T → V | T → A | I → I | I → V | I → A |
> |-|-|-|-|-|-|-|-|-|-|
> | GPT-4o | 73.0  | 58.0  | 69.5  | 52.5  | 56.0  | 26.0  | 49.0  | 42.0  | 68.0 |
> | Gemini-1.5-Pro   | 79.0  | 57.5  | 68.5  | 52.0  | 56.0  | 38.0  | 57.0  | 39.0  | 69.0 |
> | Gemini-2.0-Flash | 67.5  | 59.0  | 69.0  | 46.5  | 58.0  | 50.5  | 51.0  | 43.5  | 57.5 |
> |||||||||||
> | Qwen2.5-Omni-7B  | 50.5  | 42.9  | 51.5  | 36.0  | 31.5  | 39.5  | 29.5  | 40.0  | 32.5 |
> | Kirby-Omni-7B | 70.0  | 53.5  | 67.0  | 49.0  | 56.0  | 40.5  | 47.0  | 45.0  | 72.5 |
>
>
> &nbsp;
>
> > ***Q3. [Intuition on multimodal generalization]*** Could the authors elaborate more on the intuition of generalizing from textual reasoning data to multimodal judge capability?
>
> ***A3.*** Thank you for the question. Our intuition is motivated by two key lines of recent research:
>
> 1. It has been shown that when multilingual LLMs are pretrained to encode shared representations across languages, fine-tuning on a specific task in one language often leads to **improved performance on the same task in other languages**, even without additional data. This demonstrates the potential for knowledge transfer and skills to generalize across different domains when the underlying representations are unified.
>
> 2. Works such as s1 (Meunnighoff et al., 2025) and LIMO (Ye et al., 2025) have shown that pretrained LLMs embed rich knowledge and reasoning capabilities, which can be **"elicited" with only a small amount of additional reasoning supervision**. This pattern has also been observed in preference alignment research, such as LIMA (Zhou et al., 2023), where even a small set of high-quality instructions can unlock strong task performance in pretrained models.
>
> Based on these findings, our core intuition is that a well-pretrained multimodal LLM already possesses unified cross-modal representations and general reasoning ability. Therefore, with only a small amount of high-quality textual reasoning data, we are able to evoke a model to elicit multimodal reasoning-based judgment capabilities, without the need for large-scale, modality-specific annotations. Our work empirically verifies this hypothesis.
>
> &nbsp;
>
> > ***Q4. [Question on curated training dataset]*** How was the seed training data curated? Could the authors provide some concrete examples of the data? Given that the data is mainly synthetic, how do you ensure its high quality and avoid issues such as model collapse?
>
> ***A4.*** Thank you for your question. Our seed training data was curated by applying a set of quality criteria, which we systematically investigated through various ablation studies ***(see Figure 2 and Section 2.2)***. These criteria proved effective for producing high-quality synthetic data without suffering from mode collapse in our experiments. To further address concerns about possible collapse or forgetting, we additionally evaluated our models on visual question answering (VQA) tasks unrelated to judgment, specifically TextVQA and OK-VQA. As shown below, Kirby-VL-7B exhibits only a slight decrease on TextVQA and even outperforms the base model on OK-VQA.
>
> | Model | TextVQA | OK-VQA |
> |-|-|-|
> | Qwen2.5-VL-7B   | 80.80   | 67.46  |
> | Kirby-VL-7B     | 80.31   | 72.10  |
>
> These results suggest that our well-structured methodology does not cause model collapse or catastrophic forgetting. Also, we want to highlight that we have included the curated seed data in the supplementary material for your reference.

---

### Official Review · Reviewer_zegN · 2025-06-24

**Clarity:** 3
**Significance:** 3
**Originality:** 3
**Rating:** 5
**Confidence:** 4

**Summary:**

The paper introduces a new multimodal judge model, Kirby-Judge, designed to evaluate the performance of large multimodal models across a broad range of tasks. The proposed approach demonstrates strong robustness in handling multiple modalities, even when trained with minimal textual reasoning data. Experimental results show that Kirby-Judge outperforms state-of-the-art judge models that rely heavily on modality-specific training data, particularly in tasks such as image understanding, image editing, video generation and audio understanding. Additionally, the authors present an effective case study in which their approach is successfully adapted to evaluate multimodal models for molecular modeling.

**Questions:**

- Could the authors comment on how their model would compare against the recently proposed MLLM-as-a-Judge models such as InternLM-XComposer2.5-Reward, CAREVL, and JudgeAnything, or provide preliminary results if available?
- Have the authors considered evaluating Kirby-Judge as a response selector as part of an inference-time scaling strategy?
- Could video understanding assessments be carried out using the JudgeAnything benchmark?

**Ethical Concerns:**

["NO or VERY MINOR ethics concerns only"]

**Final Justification:**

The authors provided detailed responses to my earlier comments. I appreciate the clarifications and the additional results the authors provided, which highlight the contribution of the paper better.

**Limitations:**

The authors have adequately discussed the limitations of their approach.

**Paper Formatting Concerns:**

I did not observe any major formatting issues in the paper.

**Quality:**

3

**Strengths And Weaknesses:**

Strengths

- The paper presents Kirby-Judge, a novel multimodal judge model designed to evaluate large multimodal models across a diverse range of tasks, which addresses a critical need in the field.
- The proposed approach is robust to different input modalities and requires only minimal textual reasoning data for training, making it broadly applicable and data-efficient.
- Experimental evaluations demonstrate that Kirby-Judge, in general, outperforms existing judge models trained with extensive modality-specific data, across a diverse  set of tasks.
- The method significantly reduces the reliance on large amounts of specialized training data, enhancing its practicality and scalability in real-world applications.
- The paper includes a well-executed case study showing the adaptability of Kirby-Judge to the molecular modeling domain, further illustrating the model’s flexibility and broader applicability.

Weaknesses

- The image understanding capabilities of Kirby-Judge are evaluated using the MLLM-as-a-Judge and VL-RewardBench benchmarks. However, more recent and comprehensive benchmarks such as MM-RewardBench (Yasunaga et al., 2025), TaskAnything, and JudgeAnything (Pu et al., 2025) are not considered in the analysis.
    - Yasunaga et al., Multimodal RewardBench: Holistic Evaluation of Reward Models for Vision Language Models, arXiv preprint arXiv:2502.14191, 2025
    - Pu et al., Judge Anything: MLLM as a Judge Across Any Modality, arXiv preprint arXiv:2503.17489, 2025
- The video understanding capabilities of Kirby-Judge are not evaluated. For this, the authors could consider benchmarks like JudgeAnything (Pu et al., 2025), which support multi-modality assessment, including video.
- The evaluation of video generation and image editing capabilities is limited to the GenAI-Bench benchmark. A more comprehensive evaluation could include the recently proposed Q-Eval-100K (Zhang et al., 2025) and Video-Bench (Han et al., 2025).
    - Zhang et al., Q-Eval-100K: Evaluating Visual Quality and Alignment Level for Text-to-Vision Content, CVPR 2025
    - Han et al., Video-Bench: Human-Aligned Video Generation Benchmark, CVPR 2025
- Several recently proposed MLLM-as-a-Judge models such as InternLM-XComposer2.5-Reward (Zang et al., 2025), CAREVL (Dai et al., 2025), and JudgeAnything (Pu et al., 2025) are not included in the experimental comparison, which limits the assessment of Kirby-Judge's competitiveness.
    - Zang et al., Internlm-xcomposer2.5-reward: A simple yet effective multi-modal reward model, arXiv preprint arXiv:2501.12368, 2025
    - Dai et al., From captions to rewards (CAREVL): Leveraging large language model experts for enhanced reward modeling in large vision-language models, arXiv preprint arXiv:2503.06260, 2025
- The paper demonstrates that classical inference-time scaling techniques, such as majority voting over multiple reasoning paths, can further improve Kirby-Judge's performance. However, it does not explore whether Kirby-Judge itself can be used as a tool for inference-time scaling, e.g., selecting the best response from a set of diverse outputs generated by an MLLM. Evaluating Kirby-Judge in this context would provide valuable insight into its practical utility as a response selector.

---

> ### Author Rebuttal · Authors · 2025-07-29
>
> We thank the reviewer for their constructive feedback. Below we address each concern in detail. We look forward to any further discussions.
>
>
> >***Q1. [More recent benchmarks on image/video understanding]*** Have authors evaluated Kirby-Judge on newer image and video understanding benchmarks like MM-RewardBench or JudgeAnything to more comprehensively validate its performance?
>
> ***A1.*** Thank you for your suggestion. In response, we have extended our evaluation to include results on two recent and comprehensive MLLM evaluation benchmarks: Multimodal RewardBench (Yasunaga et al., 2025), which focuses on image understanding, and JudgeAnything (Pu et al., 2025), which covers wide range of any-to-any tasks. JudgeAnything includes not only image/video/audio understanding (I → T, V → T, A → T) but also image/video/audio generation (T → I, T → V, T → A), image editing (I → I), image-to-video (I → V), and image-to-audio (I → A) tasks. The results are summarized in the table below.
>
>
> **Multimodal RewardBench**:
>
> | Model | Overall | Correctness | Preference | Knowledge | Math  | Coding | Safety | VQA   |
> |-|-|-|-|-|-|-|-|-|
> | GPT-4o | 0.715 | 0.626 | 0.690 | 0.720 | 0.676 | 0.621 | 0.748 | 0.872 |
> | Llama-3.2-90B-Vision-Instruct | 0.624 | 0.600 | 0.684 | 0.612 | 0.563 | 0.531 | 0.520 | 0.771 |
> | Llava-1.5-13B | 0.489 | 0.533 | 0.552 | 0.505 | 0.535 | 0.493 | 0.201 | 0.518 |
> ||||||||||
> | Kirby-VL-7B | 0.686 | 0.620 | 0.618 | 0.630 | 0.677 | 0.550 | 0.732 | 0.837 |
> | Kirby-Omni-7B | 0.631 | 0.616 | 0.612 | 0.657 | 0.625 | 0.582 | 0.362 | 0.777 |
>
> **JudgeAnything**:
>
> | Model | I → T | V → T | A → T | T → I | T → V | T → A | I → I | I → V | I → A |
> |-|-|-|-|-|-|-|-|-|-|
> | GPT-4o | 73.0  | 58.0  | 69.5  | 52.5  | 56.0  | 26.0  | 49.0  | 42.0  | 68.0 |
> | Gemini-1.5-Pro   | 79.0  | 57.5  | 68.5  | 52.0  | 56.0  | 38.0  | 57.0  | 39.0  | 69.0 |
> | Gemini-2.0-Flash | 67.5  | 59.0  | 69.0  | 46.5  | 58.0  | 50.5  | 51.0  | 43.5  | 57.5 |
> |||||||||||
> | Qwen2.5-Omni-7B  | 50.5  | 42.9  | 51.5  | 36.0  | 31.5  | 39.5  | 29.5  | 40.0  | 32.5 |
> | Kirby-Omni-7B | 70.0  | 53.5  | 67.0  | 49.0  | 56.0  | 40.5  | 47.0  | 45.0  | 72.5 |
>
> Across both benchmarks, Kirby-Judge achieves comparable performance to proprietary models while consistently outperforming existing open-source baselines. However, we would like to emphasize that our ultimate goal is not to surpass proprietary models on conventional benchmarks, but to enable capabilities that they currently lack. Specifically, Kirby-Judge is designed to be adaptable to new, underexplored domains, such as molecular tasks, where existing models cannot be directly applied—which we detail further in the following response (***A4***).
>
> &nbsp;
>
>
> >***Q2. [More recent benchmarks on video generation]*** Have authors evaluated Kirby-Judge on newer benchmarks like Q-Eval-100K and Video-Bench for video generation and image editing tasks?
>
> ***A2.*** Thank you for your suggestion. We carefully considered the recent benchmarks you mentioned. However, to the best of our knowledge, the official GitHub (or Hugging Face) repository for Q-Eval-100K does not provide access to their test split, which is necessary for a fair and standardized evaluation. For Video-Bench, their evaluation framework involves using both MLLMs and LLMs simultaneously, which is not directly compatible with our evaluation protocol.
>
> Instead, the JudgeAnything results (as shown in ***A1***) show promising performances of Kirby-Omni-7B in text-to-video (T → V) and image-to-video (I → V) generation tasks. We currently believe GenAI-Bench (Table 3 in the paper) is the most widely used and representative benchmark for video generation and image editing. Our results show that Kirby-Judge is effective across a wide range of visual tasks, including video and image generation, image understanding, image editing, audio generation, and molecule understanding.
>
>
> &nbsp;
>
>
>
> >***Q3. [More recent baselines]*** Did authors consider including recently proposed MLLM-as-a-Judge models, such as InternLM-XComposer2.5-Reward or CAREVL, in your experimental comparison to more thoroughly assess the competitiveness of Kirby-Judge?
>
> ***A3.*** Thank you for the suggestion. Recent models like CaReVL and InternLM-XComposer2.5-Reward (IXC-2.5-Reward) are primarily designed only for pairwise comparison and trained on large-scale visual-language pairs, whereas Kirby-Judge is built to generalize across broader evaluation formats, including single-score grading and batchwise ranking, trained with ~1K text data.
>
> Importantly, Kirby-Judge demonstrates strong generalization not only in evaluation format, but also across domains. While many existing judge models are limited to vision-language tasks, our work uniquely extends to molecular generation (potentially to other modalities like 3D space), where curated preference data is scarce. Despite these differences, Kirby-Judge remains competitive even when evaluated against recent models, as presented in our results below.
>
> | Model | VL-RewardBench (General) | VL-RewardBench (Hallucination) | VL-RewardBench (Reasoning) | MLLM-as-a-Judge (Pair w Tie, Ave.) | MLLM-as-a-Judge (Pair wo Tie, Ave.) |
> |-|-|-|-|-|-|
> | Qwen2-VL-7B | 31.6 | 19.1 | 51.1 | 0.365 | 0.329 |
> | IXC-2.5-Reward | 84.7 | 62.5 | 62.9 | - | - |
> | CaReVL (Qwen2-VL-based) | 74.2 | 66.3 | 56.3 | 0.538 | 0.677 |
> | Kirby-VL (Qwen2-VL-based) | 42.5 | 31.8 | 57.2 | 0.524 | 0.625 |
>
> &nbsp;
>
> >***Q4. [Practical utility of Kirby-Judge for response selection]*** Have you evaluated the practical utility of Kirby-Judge as a response selector at inference time, for instance by selecting the best answer from diverse model outputs?
>
> ***A4.*** As described in ***Section 4 (Broader Impact) and Figure 4***, we have already demonstrated the use of Kirby-Judge as a best-of-N selector for inference-time scaling, particularly in the molecular domain. This shows that Kirby-Judge can effectively select the best response from a set of diverse outputs in practical scenarios. Furthermore, we demonstrated that Kirby-Judge can also serve as an off-the-shelf reward model for DPO training, where Kirby-Mol-Llama effectively guides Mol-Llama's preference optimization, highlighting its applicability to downstream training.

---

> > ### Comment · Reviewer_zegN · 2025-08-03
> >
> > Thank you for your detailed responses. I appreciate the clarifications and the additional results you provided.

---

### Official Review · Reviewer_1xgQ · 2025-06-30

**Clarity:** 3
**Significance:** 3
**Originality:** 3
**Rating:** 5
**Confidence:** 3

**Summary:**

This work proposes Kirby-Judge, a multimodal judge model for automatic evaluation of image, videos, audio, and even molecules. The model is fine-tuned from Qwen2.5-VL/Omni-7B on a small scale (1K) of text-only data for LLM-as-judge data with high-quality reasoning steps. The hypothesis is that by learning from structured, reasoning-rich text data, the model can generalize to other modalities that the model originally can understand. Despite the minimal training cost, experiments show that Kirby-Judge can produce effective judging results aligned with human preferences.

**Questions:**

Please check the Weaknesses described above.

**Ethical Concerns:**

["NO or VERY MINOR ethics concerns only"]

**Final Justification:**

The previous concerns are addressed in the response. I will keep the positive rating.

**Limitations:**

The authors have discussed the limitations in the supplementary material. Another minor limitation is that this work does not show whether this method can further scaled to larger models (e.g., 32B).

**Quality:**

3

**Strengths And Weaknesses:**

## Strengths
1. The training (fine-tuning) cost is very affordable.

2. The idea of transferring knowledge from text-only data to other modalities for LLM-as-judge is interesting.

3. The comprehensive experiment results show promising performance compared with the best open-source and closed-source models.

## Weaknesses

1. In the main Tables 1 and 2, the base models' (Qwen2.5-VL-7B, Qwen2.5-Omni-7B) performance should be included to clearly understand the relative improvement brought by Kirby-Judge.

2. In Figure 6, the standard deviation of multiple runs seems large. How does it compare with baseline models? Some more analysis on the model prediction stability/robustness could be helpful.

3. As acknowledged in the supplementary material, the success of Kirby-Judge relies on the capabilities of the underlying multimodal LLM. If the base model does not have strong reasoning capabilities already, it would be hard to develop a judge model with limited fine-tuning data. In other words, the performance of Kirby-Judge may be heavily dependent on the base models (Qwen2.5-VL/Omni-7B), especially their base reasoning capabilities. Considering reasoning LLMs are being actively developed in the open-source community, this limitation is rather minor, but worth some investigation (e.g., training weaker models with the same data as Kirby-Judge).

---

> ### Author Rebuttal · Authors · 2025-07-30
>
> We appreciate your constructive comments. We have rephrased the question for ease of reference and provide our corresponding responses below. We look forward to any further discussions.
>
> > ***Q1. [Comparison with base model performances]*** Could the authors include the results of the base models (Qwen2.5-VL-7B, Qwen2.5-Omni-7B) in the main tables to clearly show how much improvement is achieved by Kirby-Judge?
>
> ***A1.*** Thank you for your comment. As shown in the table below, we directly compared our Kirby-Judge models (Kirby-VL-7B and Kirby-Omni-7B) to the base instruction-tuned models (Qwen2.5-VL-7B and Qwen2.5-Omni-7B) across multiple benchmarks. With only a small amount of text-only data, Kirby-Judge achieves substantial performance improvements over the base models—for example, up to +32.38 on VL-RewardBench (Reasoning) and +14.85 on GenAI-Bench (Edition). This demonstrates that our fine-tuning approach is highly effective compared to simply prompting the instruction-tuned base models.
>
> | Model | MLLM-as-a-judge (Score, Ave.) | MLLM-as-a-judge (Pair w Tie, Ave.) | MLLM-as-a-judge (Pair w.o Tie, Ave.) | MLLM-as-a-judge (Batch, Ave.) $\downarrow$ | VL-RewardBench (General) | VL-RewardBench (Hallu.) | VL-RewardBench (Reason.) | GenAI-Bench (Image) | GenAI-Bench (Edition) | GenAI-Bench (Video) |
> |-|-|-|-|-|-|-|-|-|-|-|
> | Qwen2.5-VL-7B | 0.165 | 0.423 | 0.425 | 0.585 | 37.65 | 33.11 | 48.19 | 31.93 | 38.63 | 37.61 |
> | Kirby-VL-7B | 0.332 | 0.538 | 0.655 | 0.426 | 46.11 | 43.39 | 62.87 | 43.32 | 47.41 | 44.78 |
> | Δ (Kirby - Qwen) | +0.167 | +0.115 | +0.230 | -0.159 | +8.46 | +10.28 | +14.68 | +11.39 | +8.78 | +7.17 |
> | | | | | | | | | | | |
> | Qwen2.5-Omni-7B | 0.072 | 0.471 | 0.526 | 0.576 | 32.60 | 18.30 | 28.70 | 34.87 | 31.88 | 38.45 |
> | Kirby-Omni-7B | 0.306 | 0.532 | 0.650 | 0.425 | 47.01 | 42.72 | 61.08 | 38.15 | 46.73 | 37.10 |
> | Δ (Kirby - Qwen) | +0.234 | +0.061 | +0.124 | -0.151 | +14.41 | +24.42 | +32.38 | +3.28 | +14.85 | -1.35 |
>
> &nbsp;
>
> > ***Q2. [Large standard deviation of Kirby-Omni-7B]*** In Figure 6, the standard deviation of multiple runs (especially for Kirby-Omni-7B) appears to be large. How does this compare to the baseline models? Some further analysis on model prediction stability or robustness would be helpful.
>
> ***A2.*** Thank you for your question. The table below reports the average (standard deviation) of model accuracy across multiple runs, where $k$ is the number of reasoning paths used for majority voting. We observe that the baseline instruction-tuned Omni model (Qwen2.5-Omni-7B) already exhibits high variance, and this characteristic is inherited by Kirby-Omni-7B. In contrast, both the baseline VL model and Kirby-VL-7B show much lower standard deviation, indicating greater stability. Importantly, for both VL and Omni variants, the Kirby-Judge model either matches or reduces the standard deviation relative to its base model at each value of $k$. These findings suggest that prediction stability (as measured by standard deviation) is mainly a property of the base model. Our method does not introduce additional instability and, in some cases, even improves evaluation consistency compared to the baseline.
>
> | Model | $k=1$ | $k=3$ | $k=5$ | $k=7$ | $k=9$ |
> |-|-|-|-|-|-|
> | Qwen2.5-VL-7B      | 22.53 (4.95)  | 42.73 (2.91)  | 47.79 (1.99)  | 51.20 (1.92)  | 53.29 (1.75)  |
> | Kirby-VL-7B        | 55.39 (1.36)  | 57.31 (1.22)  | 58.32 (0.71)  | 59.52 (0.80)  | 60.12 (0.61)  |
> | Qwen2.5-Omni-7B    |  9.48 (4.79)  | 22.45 (4.63)  | 28.58 (4.22)  | 32.89 (3.33)  | 37.37 (2.21)  |
> | Kirby-Omni-7B      | 55.01 (1.79)  | 56.44 (2.14)  | 57.68 (3.14)  | 57.38 (2.91)  | 57.64 (2.03)  |
>
> &nbsp;
>
> > ***Q3. [Dependence on base model capabilities]*** As acknowledged in the supplementary material, the effectiveness of Kirby-Judge is influenced by the reasoning capacity of the backbone model. Could the authors comment on how your approach performs with weaker or smaller backbone models, and whether you conducted related experiments?
>
> ***A3.*** Thank you for your insightful question. We agree that the performance of Kirby-Judge is closely tied to the multimodal understanding and reasoning ability of its backbone MLLM. To directly assess this, we conducted additional experiments with a weaker backbone: we trained Kirby-VL-3B by fine-tuning Qwen2.5-VL-3B on our curated seed dataset and compared its performance to Kirby-VL-7B and other larger models (see ***Table 7 and Appendix C.1*** in the supplementary material). Notably, Kirby-VL-3B demonstrates strong generalization and reasoning ability, achieving better results than LLaVA-NeXT and Qwen2.5-VL-7B, but underperforms compared to our Kirby-VL-7B model.
>
> Additionally, we further experimented with another weaker backbone, Qwen2-VL-7B, in place of Qwen2.5-VL-7B. This resulted in a slight further performance drop: 0.538 → 0.524 (MLLM-as-a-Judge, pair w tie) and 46.1 → 42.5 (VL-RewardBench, General), but still highly competitive to the open-source baselines. These results confirm that while the choice of backbone affects the performance, our Kirby-Judge consistently brings significant gains over the base models, even with smaller or weaker MLLMs.
>
> Moreover, as the reviewer pointed out, reasoning-capable MLLMs are rapidly evolving. Recent models like DeepSeek-R1-Distill show promising reasoning ability even at smaller scales, and our framework is designed to benefit from these ongoing advancements. Once a stronger or more efficient backbone becomes available, our method can be seamlessly applied using only a small amount of high-quality text-based supervision.
>
> &nbsp;
>
> > ***Q4. [Larger Kirby-Judge]*** Another minor limitation is that this work does not show whether this method can be further scaled to larger models (e.g., 32B).
>
> ***A4.*** Thank you for pointing this out. We have trained Kirby-VL-32B, based on Qwen2.5-VL-32B-Instruct (see table below), and these results confirm the scalability of our framework. Although we trained the 32B model using parameter-efficient LoRA (Hu et al., 2021), which is generally less effective than full fine-tuning due to limited computational resources, it outperforms the fully fine-tuned 7B version on most evaluation benchmarks and is competitive with GPT-4o. This validates that judgment quality improves as the backbone’s reasoning capacity increases.
>
> | Model | MLLM-as-a-judge (Score, Ave.) | MLLM-as-a-judge (Pair w Tie, Ave.) | MLLM-as-a-judge (Pair w.o Tie, Ave.) | MLLM-as-a-judge (Batch, Ave.) $\downarrow$ | VL-RewardBench (General) | VL-RewardBench (Hallu.) | VL-RewardBench (Reason.) | GenAI-Bench (Image) | GenAI-Bench (Edition) | GenAI-Bench (Video) |
> |-|-|-|-|-|-|-|-|-|-|-|
> | GPT-4V/4o | 0.424 | 0.538 | 0.717 | 0.361 | 49.10 | 67.60 | 70.50 | 45.59 | 53.54 | 48.46 |
> | | | | | | | | | | |
> | Kirby-VL-3B   | 0.176 | 0.493 | 0.650 | 0.478 | 46.39 | 36.85 | 58.08 | 36.71 | 33.62 | 42.93 |
> | Kirby-VL-7B   | 0.332 | 0.538 | 0.655 | 0.426 | 46.11 | 43.39 | 62.87 | 43.32 | 47.41 | 44.78 |
> | Kirby-VL-32B (LoRA) | 0.361 | 0.587 | 0.716 | 0.432 | 54.52 | 41.92 | 64.46 | 44.32 | 53.54 | 47.33 |

---

> > ### Comment · Reviewer_1xgQ · 2025-08-02
> >
> > The authors' detailed response is greatly appreciated. The original concerns are addressed.

---

### Official Review · Reviewer_rdTF · 2025-07-04

**Clarity:** 3
**Significance:** 2
**Originality:** 2
**Rating:** 4
**Confidence:** 3

**Summary:**

The paper presents the pipeline and the model for serving as a task evaluator that is trained using only textual reasoning data that can generalize its judgment capabilities to various modalities (vision, audio, video, even molecular inputs) without additional modality-specific training. The core idea of the proposed approach is that structured explanatory rationales (specifically generated for the responses evaluation and comparison) capture general decision-making patterns that can transfer across modalities. In the experiments, the authors fine-tune a base multimodal language model (specifically, Qwen-2.5 VL/Omni, 7B) on a small curated corpus of high-quality text-only evaluation examples with rich reasoning. After this training, the KIRBY-Judge (the fine-tuned model) can judge any input, i.e. perform zero-shot evaluation on image-text tasks, audio transcripts, video descriptions, and biological data, by leveraging the base model’s multimodal inputs. Empirically, KIRBY-Judge achieves competitive or superior performance to state-of-the-art evaluators despite using orders of magnitude less training data.

**Questions:**

I have the following questions for the authors:
1. Did you evaluate the performance of the specifically prompted, but instruct-tuned multimodal models on your judgement tasks?
2. Did you experiment with the size of the dataset for finetuning, can you assess the number of textual data necessary for training without sacrificing the multimodal abilities of the initial model?

**Ethical Concerns:**

["NO or VERY MINOR ethics concerns only"]

**Final Justification:**

I have read the rebuttals from the authors for my review and other reviewers, and I am quite satisfied with the responses. One of my initial concerns was that including more textual data for training can drop the model's performance significantly; the authors provided additional evaluation, showing that this is not the case. So, I would like to increase my score and suggest accepting the paper because the topic of evaluating multimodal inputs is essential in the current domain, where benchmarks are concentrated on generative answers that require oracles to judge the responses.

Revised Score: 5 (Accept)

**Limitations:**

yes

**Quality:**

3

**Strengths And Weaknesses:**

The main strengths of the paper are as follows:

1. The paper analyzes and empirically validates an important idea: that it is possible to effectively train models using small-scale datasets composed solely of high-quality, text-only reasoning examples. The main results suggest that when collecting modality-specific annotations is challenging, such datasets can still yield performance comparable to that of models trained on large-scale, modality-specific data.
2. The specific molecular case study strengthens the contribution by demonstrating the method’s applicability to specific domain data. This shows that the proposed approach can generalize beyond conventional multimodal settings.
3. The paper provides extensive experimental evaluation across a diverse set of benchmarks and modalities, which reinforces the robustness and reliability of the proposed method.

The main weaknesses of the paper are as follows:

1. Despite the thorough comparisons with existing models, it would be valuable to include additional ablation studies to better isolate the effects of the proposed fine-tuning approach. First, it would be useful to evaluate the performance of the base models (e.g., Qwen-VL and Qwen-Omni) without any fine-tuning on the text-only dataset. This could clarify whether the instruction-following capabilities of the base models alone are sufficient for judgment tasks, or if the fine-tuning step is indeed necessary.
Second, assessing the performance gap on standard vision-language benchmarks before and after text-only fine-tuning would help determine whether this fine-tuning affects the model’s inherent multimodal understanding capabilities (either positively or negatively).
2. An additional ablation study examining the effect of dataset size would also be valuable. Specifically, it would be helpful to understand how much text-only data is actually needed to achieve strong generalization performance without sacrificing the multimodal abilities of the initial checkpoint.

---

> ### Author Rebuttal · Authors · 2025-07-29
>
> Thank you for your insightful feedback. We have rephrased your comments for simpler reference and have included our respective responses. We look forward to any further discussions.
>
> >***Q1. [Comparison with base model performances]*** Did authors compare the performance of your base models and fine-tuned models on judgment tasks, and did you assess how text-only fine-tuning affects their multimodal understanding on standard vision-language benchmarks?
>
> ***A1.*** Thank you for your comment. As shown in the table below, we directly compared our Kirby-Judge models (Kirby-VL-7B and Kirby-Omni-7B) to the base instruction-tuned models (Qwen2.5-VL-7B and Qwen2.5-Omni-7B) across multiple benchmarks. With only a small amount of text-only data, Kirby-Judge achieves substantial performance improvements over the base models—for example, up to +32.38 on VL-RewardBench (Reasoning) and +14.85 on GenAI-Bench (Edition). This demonstrates that our fine-tuning approach is highly effective compared to simply prompting the instruction-tuned base models.
>
> | Model | MLLM-as-a-judge (Score, Ave.) | MLLM-as-a-judge (Pair w Tie, Ave.) | MLLM-as-a-judge (Pair w.o Tie, Ave.) | MLLM-as-a-judge (Batch, Ave.) $\downarrow$ | VL-RewardBench (General) | VL-RewardBench (Hallu.) | VL-RewardBench (Reason.) | GenAI-Bench (Image) | GenAI-Bench (Edition) | GenAI-Bench (Video) |
> |-|-|-|-|-|-|-|-|-|-|-|
> | Qwen2.5-VL-7B | 0.165 | 0.423 | 0.425 | 0.585 | 37.65 | 33.11 | 48.19 | 31.93 | 38.63 | 37.61 |
> | Kirby-VL-7B | 0.332 | 0.538 | 0.655 | 0.426 | 46.11 | 43.39 | 62.87 | 43.32 | 47.41 | 44.78 |
> | Δ (Kirby - Qwen) | +0.167 | +0.115 | +0.230 | -0.159 | +8.46 | +10.28 | +14.68 | +11.39 | +8.78 | +7.17 |
> | | | | | | | | | | | |
> | Qwen2.5-Omni-7B | 0.072 | 0.471 | 0.526 | 0.576 | 32.60 | 18.30 | 28.70 | 34.87 | 31.88 | 38.45 |
> | Kirby-Omni-7B | 0.306 | 0.532 | 0.650 | 0.425 | 47.01 | 42.72 | 61.08 | 38.15 | 46.73 | 37.10 |
> | Δ (Kirby - Qwen) | +0.234 | +0.061 | +0.124 | -0.151 | +14.41 | +24.42 | +32.38 | +3.28 | +14.85 | -1.35 |
>
>
> To further address concerns about potential catastrophic forgetting or degradation of multimodal abilities, we also evaluated our models on standard visual question answering (VQA) tasks that are unrelated to judgment, specifically TextVQA (Signh et al., 2019) and OK-VQA (Marino et al., 2019). As shown below, Kirby-VL-7B exhibits only a slight decrease on TextVQA and even outperforms the base model on OK-VQA. These results suggest that our well-structured methodology preserves the multimodal capabilities of the base model and does not cause forgetting.
>
> | Model | TextVQA | OK-VQA |
> |-|-|-|
> | Qwen2.5-VL-7B   | 80.80   | 67.46  |
> | Kirby-VL-7B     | 80.31   | 72.10  |
>
> In summary, our experiments demonstrate that Kirby-Judge achieves significant improvements over the base models on judgment tasks, while largely maintaining (and in some cases improving) performance on non-judgment multimodal benchmarks.
>
> &nbsp;
>
> >***Q2. [Effect of dataset size]*** How does the amount of text-only data used for fine-tuning affect generalization and multimodal performance? Did you perform ablation studies on this?
>
> ***A2.*** Thank you for the comment. As shown in ***Section 2.2 and Figure 2(b)***, we have provided an ablation study on the effect of dataset size for both text-only (PandaLM, JudgeLM) and vision-language (VL-RewardBench, MJ-Bench) judge evaluations. We find that text-only tasks benefit from larger datasets, showing continuous improvement as the number of samples increases. In contrast, for vision-language tasks, performance improves up to a certain threshold (1K), but adding more data beyond that point results in a slight and gradual performance drop, likely due to catastrophic forgetting. These results highlight the importance of careful dataset size selection, and we have designed our methodology with these findings in mind.

---

> > ### Comment · Reviewer_rdTF · 2025-08-07
> > **Official Comment by Reviewer rdTF**
> >
> > I would like to thank the authors for the detailed response and additional experiments. It is specifically interesting that the proposed training setup didn't drop the multimodal ability of the aligned model. I also investigated the supplementary materials with the appendix that provides the system prompts used for the evaluation. My only question to the authors is whether the same system prompts were used for the original instruct-tuned models to obtain the results in the table in your rebuttal? Overall, all my concerns are covered by the authors' response.

---

> > > ### Author Response · Authors · 2025-08-08
> > >
> > > We appreciate the reviewer’s positive feedback and are glad that our response and additional experiments have addressed your concerns. Regarding your question: yes, we confirm that the same system prompts were used for the original instruct-tuned models when producing the results shown in the table in our rebuttal. This ensured a fair and consistent evaluation across all models.

---

### Note · Authors · 2025-08-12

Dear Area Chairs and Reviewers,

We are grateful for your insightful feedback and the opportunity to engage in a productive discussion. We are pleased to confirm that **all reviewer concerns have been fully addressed** through our detailed rebuttals and additional experiments. We have successfully resolved the raised questions/suggestions including: ***clarification on the dataset curation***, ***comparison with base models***, ***additional benchmarks***, and ***scaling Kirby-Judge to 32B***.

Our paper introduces **Kirby-Judge**, a novel and efficient method for multimodal evaluation. By training on a small, high-quality dataset of textual reasoning, our model is able to effectively generalize across diverse modalities—from images and videos to molecular structures—without the need for costly, modality-specific data.

A key strength of our approach is its practicality. Kirby-Judge achieves performance that is competitive with, and often superior to, state-of-the-art models that require significantly more resources. We believe this makes our method a crucial tool for ***test-time scaling in resource-constrained environments***. This adaptability has been demonstrated in molecules and is expected to be applicable in the future to domains such as Video-3D [1], medical [2], and various natural sciences [3].

We are confident that our work offers a significant and practical contribution, and we thank you for your time and consideration.

Best regards, \
Authors of Submission 16251

**References**

[1] Zheng et al., "Video-3D LLM: Learning Position-Aware Video Representation for 3D Scene Understanding." CVPR. 2025 \
[2] Google Research and Google DeepMind. "MedGemma Technical Report." arXiv 2507.05201., 2025 \
[3] Microsoft Research. "Natural Language Model: Deciphering the Language of Nature for Scientific Discovery." arXiv 2502.07527., 2025

---

### Decision · Program_Chairs · 2025-09-17

**Decision:**

Accept (poster)

**Comment:**

This paper was reviewed by four knowledgeable reviewers. The reviewers raised concerns about ablation studies w.r.t base models [rdTF, 1xgQ], the effect of dataset size [rdTF], the performance on understanding tasks [rdTF], the selected benchmarks [zegN, VVyF] and baselines considered [VVyF, zegN]. The rebuttal adequately addressed the reviewers' concerns and introduced the requested ablation studies, provided evaluations on VQA tasks, analyzed the prediction robustness of the models, and showed results on weaker backbones and larger models. The rebuttal also incorporated the more recent benchmarks and comparisons suggested by the reviewers. After rebuttal and discussion, there is a clear consensus to accept the paper. The AC appreciates the efforts put in the rebuttal, which substantially strengthen the contribution, and agrees with the reviewers' recommendation. Therefore, the AC recommends to accept.